



# The roles of volatile organic compound deposition and oxidation mechanisms in determining secondary organic aerosol production: A global perspective using the UKCA chemistry-climate model (vn8.4)

Jamie. M. Kelly[1], Ruth M. Doherty[1], Fiona. M. O'Connor[2], Graham W. Mann[3], Hugh Coe[4], Dantong Liu[4]

[1]School of GeoSciences, The University of Edinburgh, U.K
[2]Met Office Hadley Centre, Exeter, U.K
[3]National Centre for Atmospheric Science, School of Earth and Environment, University of Leeds, Leeds, U.K
[4]Centre for Atmospheric Sciences, School of Earth and Environmental Sciences, University of Manchester, Manchester, UK,
M13 9PL

*Correspondence to*: Jamie Kelly (j.kelly-16@sms.ed.ac.uk)

**Abstract.** The representation of volatile organic compound (VOC) deposition and oxidation mechanisms in the context of secondary organic aerosol (SOA) formation are developed in the United Kingdom Chemistry and Aerosol (UKCA) chemistry-

climate model. Impacts of these developments on both the global SOA budget and model agreement with observations is quantified. Firstly, global model simulations were performed with varying VOC dry deposition and wet deposition fluxes. Including VOC dry deposition reduces the global annual-total SOA production rate by 2 - 32 %, with the range reflecting uncertainties in surface resistances. Including VOC wet deposition reduces the global annual-total SOA production rate by 15 % and is relatively insensitive to changes in effective Henry's Law coefficients. With precursor deposition, simulated SOA

concentrations are lower than observed, with a normalised mean bias (NMB) of -51%. Hence, including SOA precursor deposition worsens model agreement with observations even further (NMB = -66 %). Secondly, for the anthropogenic and biomass burning VOC precursors of SOA (VOC$_{ANT/BB}$), model simulations were performed varying: a) the parent hydrocarbon reactivity, b) the number of reaction intermediates, and c) accounting for differences in volatility between oxidation products from various pathways. These changes were compared to a scheme where VOC$_{ANT/BB}$ adopts the reactivity of a monoterpene

(α-pinene), and is oxidised in a single-step mechanism with a fixed SOA yield. By using the chemical reactivity of either benzene, toluene or naphthalene for VOC$_{ANT/BB}$, the global annual-total VOC$_{ANT/BB}$ oxidation rate changes by -3, -31 or -66 %, respectively, compared to when using α-pinene. Increasing the number of reaction intermediates, by introducing a peroxy radical (RO$_2$), slightly slows the rate of SOA formation, but has no impact on the global annual-total SOA production rate. However, RO$_2$ undergoes competitive oxidation reactions, forming products with substantially different volatilities.

Accounting for the differences in product volatility between RO$_2$ oxidation pathways increases the global SOA production rate



by 153 % compared to using a single SOA yield. Overall, for relatively reactive compounds, such as toluene and naphthalene, the reduction in reactivity for $VOC_{ANT/BB}$ oxidation is outweighed by accounting for the difference in volatility of $RO_2$ products, leading to a net increase in the global annual-total SOA production rate of 85 and 145 %, respectively, and improvemtns in model agreement (NMB of -46 and 56 %, respectively). However, for benzene, the reduction in $VOC_{ANT/BB}$

oxidation is not outweighed by accounting for the difference in SOA yield pathways, leading to a small change in the global annual-total SOA production rate of -3 %, and a slight worsening of model agreement with observatiobs (NMB = -77 %). These results highlight that variations in both VOC deposition and oxidation mechanisms contribute to substantial uncertainties in the global SOA budget and model agreement with observations.

**1 Introduction**

Aerosols are detrimental to human health (WHO, 2013) and are linked to climate change (Forster and Ramaswamy, 2007). The development of air quality and climate management plans are hindered by the challenges in representing aerosol within models. Secondary organic aerosol (SOA) is formed in the atmosphere from a variety of hydrocarbons. Gas-phase production of SOA occurs by condensation of volatile organic compound (VOC) oxidation products (Odum et al., 1996;Odum et al.,

1997) and from semi-volatile and intermediate-volatility compounds (S/IVOCs) (Donahue et al., 2006;Donahue et al., 2011). Additionally, SOA formation can take place within the aqueous phase of cloud and aerosol liquid water (McNeill, 2015;Ervens, 2015). The treatment of hydrocarbon physicochemical processes within SOA schemes varies sizably across global chemistry-climate and chemical transport models, and this is reflected in both an uncertain global SOA budget and poor model agreement with observations (Tsigaridis et al., 2014).

The diversity in model treatment of SOA formation is partially due to the myriad of unique organic molecules in the atmosphere, a small fraction of which have been measured (Goldstein and Galbally, 2007). In the simplest of schemes, production of SOA is calculated as a function of emissions, hence, SOA is 'emitted' as opposed to being formed in the atmosphere (Tsigaridis et al., 2014). In cases where gas-phase oxidation of SOA precursors is treated, several simplifications are commonly made. For example, biogenic VOCs, such as isoprene and monoterpenes, are known to have multigenerational

oxidation mechanisms, but the mechanisms are often reduced to less than two reaction steps when implemented in global models (Chung and Seinfeld, 2002;Heald et al., 2011;Scott et al., 2014;Scott et al., 2015). Similarly, multigenerational oxidation mechanisms of aromatic compounds are often represented by less than two reaction steps (Tsigaridis and Kanakidou, 2003;Heald et al., 2011). Gas-phase oxidation schemes can also be simplified by grouping organic compounds together (i.e. 'lumping'). In some schemes, organic compounds are lumped according to emissions types, anthropogenic or biomass burning

(Spracklen et al., 2011;Hodzic et al., 2016) whereas in others, they are grouped according to volatility (Donahue et al., 2006;Donahue et al., 2011). By lumping organic species together, chemical ageing can be accounted for, even if the exact mechanism is not known. However, in grouping species together, molecular information is lost and therefore it is challenging



to select the appropriate reaction coefficients and SOA yields from laboratory studies (Kelly et al., 2018). In more complex SOA schemes, gas-phase oxidation is treated explicitly (Lin et al., 2012;Lin et al., 2014;Khan et al., 2017), but this method is limited to SOA precursors with relatively well-known oxidation mechanisms.

The sources and physicochemical processes of hydrocarbons included within SOA schemes also varies between

models. Examples of model diversity include the inclusion of SOA formation within the aqueous phase (Lin et al., 2014) and from S/IVOCs (Pye and Seinfeld, 2010), as well as SOA being treated as semi-volatile as opposed to non-volatile (Shrivastava et al., 2015). The treatment of dry (Bessagnet et al., 2010) and wet deposition (Knote et al., 2015) of SOA precursors is an aspects of SOA which varies from model to model. Recent field and modelling studies have provided evidence that several known SOA precursors are susceptible to deposition. For example, explicit modelling of the oxidation of terpene and aromatic

VOCs has identified extremely soluble products, with effective Henry's constants ($H_{eff}$) ranging from $10^5$ to $10^9$ M atm$^{-1}$ (Hodzic et al., 2014). This suggests efficient wet removal of SOA precursors, considering $H_{eff}$ for nitric acid ($HNO_3$) is ~2 $x10^5$ (Seinfeld and Pandis, 2006). However, the molecular-specific deposition parameters determined in field studies (Nguyen et al., 2015) can be difficult to apply to the lumped compounds used in global SOA schemes. On a global scale, some modelling studies have indicated a sensitivity of SOA to variations in precursor deposition (Henze and Seinfeld, 2006;Pye and Seinfeld,

2010;Hodzic et al., 2016). A few global modelling studies include both dry and wet deposition of SOA precursors, but the deposition parameters used vary by several orders of magnitude. For example, Shrivastava et al. (2015) use a value for $H_{eff}$ of 7 $x10^3$ M atm$^{-1}$, whereas other studies use values ranging from 1 $x10^5$ to 5.3 $x10^9$ M atm$^{-1}$ (Knote et al., 2015;Hodzic et al., 2016). In relation to dry deposition, field studies over forested regions of the USA have observed significant dry deposition of highly oxygenated VOCs (Nguyen et al., 2015). The most rigorous studies on dry deposition have only been conducted using

regional scale models. They found that dry removal of SOA precursors reduces modelled July-mean surface SOA concentrations by 20 – 40 % over Europe (Bessagnet et al., 2010), and reduces annual-average surface SOA concentrations by 46 % over the USA (Knote et al., 2015). Wet removal of SOA precursors reduces simulated annual-average surface SOA concentrations by 10 % over the USA, which reduces simulated positive biases in summertime SOA (Knote et al., 2015). However, previous studies have found that observed SOA concentrations in mid-latitude emission source regions tend to be

lower compared to SOA concentrations simulated without the inclusion of VOC deposition (Kelly et al. 2018), but noted that elsewhere the lack of measurements precluded robust conclusions.

Vegetation is estimated to release around 1000 Tg (C) of VOCs into the atmosphere annually (Guenther et al., 2006;Guenther et al., 2012). Estimates of the global annual-total SOA production rate from biogenic VOCs range from 27.6 to 97.5 Tg (SOA) a$^{-1}$, which represents 54 to 95 % of production from all sources (Farina et al., 2010;Hodzic et al., 2016).

However, other emissions, such as fossil fuel and biofuel combustion, as well as savannah and forest fires, may also be important sources of SOA. In urban environments, aromatic compounds, which are typically emitted from anthropogenic and biomass burning activities, account for 20 to 30 % of total VOC emissions (Carlton et al., 2000). Therefore, in some cities, such as Beijing (Guo et al., 2012), Shanghai (Peng et al., 2013), Guangzhou (Ding et al., 2012) and Jerusalem (Von Schneidemesser et al., 2010), SOA is primarily composed of aromatic compounds, as opposed to biogenic species. One



observationally-constrained global modelling study estimates an anthropogenically-controlled global annual-total SOA production rate of 100 Tg (SOA) a$^{-1}$, representing ~70 % of production from all sources (Spracklen et al., 2011). In other regions, SOA can be dominated by biomass burning sources (Tiitta et al., 2014). The extrapolation of observations from aircraft campaigns (Cubison et al., 2011) and environmental chamber experiments (Bruns et al., 2016) suggests a global annual-total

SOA production rate from biomass burning of 8 and 43 Tg (SOA)a$^{-1}$, respectively. Furthermore, one global-scale modelling studies predicts a global annual-total SOA production rate of 44 to 95 Tg (SOA) a$^{-1}$ from biomass burning S/IVOCs (Shrivastava et al., 2015). Therefore, the dominant sources of SOA remains largely unknown.

Until recently, laboratory-derived SOA yields have not been able to fully account for the strength of SOA production from aromatic compounds observed in field studies. On the contrary, early estimates of SOA yields from aromatic compounds,

which were conducted in relatively high nitrogen oxide (NO$_x$ = NO and NO$_2$) concentrations, range between 5 and 10 % (Odum et al., 1997;Odum et al., 1996). Consequently, the use of low SOA yields for aromatic compounds in global models results in low global annual-total SOA production rates, ranging from just 0.05 to 2.5 Tg (SOA) a$^{-1}$, which are negligible in comparison to biogenic sources (Tsigaridis and Kanakidou, 2003;Hoyle et al., 2007). However, more recent chamber studies suggest the SOA yields from aromatic compounds are strongly influenced by NO$_x$ concentrations (Hurley et al., 2001;Song et

al., 2005;Ng et al., 2007;Chan et al., 2009). For example, in agreement with early estimates (Odum et al., 1996;Odum et al., 1997), Ng et al. (2007) also observed an SOA yield from aromatic VOCs of 5 – 10 % under high-NO$_X$ conditions. However, under lower NO$_X$ concentrations, Ng et al. (2007) measured substantially higher SOA yields of 37, 30 and 36 % for benzene (C$_6$H$_6$), toluene (C$_7$H$_8$)  and xylene (C$_8$H$_{10}$), respectively. Similarly, under low-NO$_X$ conditions, Chan et al. (2009) observed an SOA yield of 73 % from naphthalene (C$_{10}$H$_8$).

The exact mechanism describing aromatic oxidation is not yet fully understood, despite considerable progress to date (Kautzman et al., 2010;Li et al., 2016;Al-Naiema and Stone, 2017;Li et al., 2017b;Schwantes et al., 2017). As aromatic oxidation is initiated by the hydroxyl radical (OH), the influence of NO$_x$ on SOA production is probably due to reaction of NO with second or later generation oxidation products. Oxidation of the parent aromatic hydrocarbon by OH is followed by addition of molecular oxygen (O$_2$) and isomerization, forming a bicyclic peroxy radical, RO$_2$ (Johnson et al., 2004;Koch et al.,

2007). Under high-NO$_x$ conditions, the peroxy radical reacts with the nitric oxide radical (NO) to form semi-volatile products, whereas, under low-NOx conditions, the peroxy radical reacts with the hydroperoxyl radical (HO$_2$) to form non-volatile products (Ng et al., 2007).  Hence, due to the difference in volatility of products, the RO$_2$+HO$_2$ yields a greater mass of SOA compared to the RO$_2$+NO pathway. Water vapour may also be involved in the gas-phase oxidation of aromatic compounds (Hinks et al., 2018). However, as both positive (White et al., 2014) and negative (Cocker et al., 2001) correlations between

aromatic SOA yields and relative humidity have been observed in chamber studies, the role of water vapour in aromatic oxidation is not yet clear. The exact mechanism describing aromatic oxidation may not be fully understood but the observed influence of NOx on SOA yields suggests that simulating SOA production from aromatic compounds necessitates multigenerational oxidation mechanisms, with SOA yields responding to oxidant availability.



The peroxy radical reaction intermediate, together with competitive NO and $HO_2$ reactions with varying SOA yields, has been applied to several different SOA schemes. Benzene, toluene and xylene have been incorporated into both global (Henze et al., 2008;Heald et al., 2011) and regional scale (Li et al., 2017a) models. Henze et al. (2008) applied the laboratory-derived yields from Ng et al. (2007) to aromatic compounds (16 Tg (VOC) $a^{-1}$), which resulted in a global annual-total SOA production rate of 4 Tg (SOA) $a^{-1}$, with 61% of SOA being produced via the $RO_2+HO_2$ pathway. Peroxy radical chemistry has also been applied to IVOCs, which are a mixture of species emitted from both anthropogenic and biomass burning. Pye and Seinfeld (2010) applied the laboratory-derived yields from Chan et al. (2009) to IVOCs (18 Tg (VOC) $a^{-1}$), which resulted in global annual-total SOA production rate of 5 Tg (SOA) $a^{-1}$, with 75% of SOA being produced via the $RO_2+HO_2$ pathway. Despite peroxy radical chemistry being included in some SOA schemes, the influence on the global SOA budget and model agreement with observations has not been quantified.

The objective of this study is to further develop the SOA scheme within a chemistry-climate model, the United Kingdom Chemistry and Aerosol (UKCA) model. Firstly, the model is updated to include the wet and dry deposition of SOA precursors. Secondly, the mechanism describing SOA formation from anthropogenic and biomass burning VOCs is updated to account for the influence of NOx on SOA yields. Several simulations are conducted to test the sensitivity of SOA to both precursor deposition and oxidation mechanisms. The impact of these model developments on SOA is assessed through a comprehensive comparison with available observations. The paper is organised as follows. The global chemistry-climate model used in this study is described in Section 2; this section also includes a description of the model developments applied to the SOA scheme. Observations used to evaluate the model are discussed in Section 3. Next, the influence of precursor deposition on SOA is investigated (Section 4). In Section 5, the sensitivity of modelled SOA to oxidation mechanisms and VOC reactivity is explored. Concluding remarks and further work are discussed in Section 6.

## 2 Chemistry-climate model description

In this section, the model is briefly described. This begins with a brief description of the default configuration, followed by the model developments made in this study. The chemistry-climate model used in this study is the United Kingdom Chemistry and Aerosol (UKCA) model (Morgenstern et al., 2009; Mann et al., 2010; O'Connor et al. 2014) which is coupled to the Global Atmosphere 4.0 (GA4.0) configuration (Walters et al., 2014) of the Hadley Centre Global Environmental Model (Hewitt et al., 2011) version 3 (HadGEM3). The atmosphere-only configuration with prescribed sea surface temperature and sea ice fields based on 1995-2004 reanalyses data (Reynolds et al., 2007) was used. The model was run at a horizontal resolution of N96 (1.875° longitude by 1.25° latitude) with 85 terrain-following hybrid-height levels distributed from the surface to 85 km. Horizontal winds and temperature in the model were nudged towards ERA-Interim reanalyses for the 1999-2000 period (Dee et al., 2011) using a Newtonian relaxation technique with a relaxation time constant of 6 hours (Telford et



al., 2008). There was no feedback from the chemistry or aerosols onto the dynamics of the model; this ensured identical meteorology across all simulations so that differences in SOA were solely due to differences in precursor oxidation mechanisms and deposition.

## 2. 1 Gaseous chemistry (UKCA)

The United Kingdom Chemistry and Aerosol (UKCA) model used in this study combines the "TropIsop" tropospheric chemistry scheme from O'Connor et al. (2014) with the stratospheric chemistry scheme from Morgenstern et al. (2009). There are 75 species with 285 reactions. This includes odd oxygen ($O_x$), nitrogen ($NO_y$), hydrogen ($HO_x = OH + HO_2$), and carbon monoxide (CO). Explicit hydrocarbons included are methane, ethane, propane, isoprene and monoterpene. Isoprene oxidation follows the Mainz Isoprene Mechanism (Poschl et al., 2000) which is described in detail in O'Connor et al. (2014). In addition to the aforementioned explicit hydrocarbons, two additional non-explicit VOCs are included; $VOC_{ANT}$ and $VOC_{BB}$ are lumped compounds representing anthropogenic and biomass burning VOCs, respectively. Together, isoprene, monoterpene, $VOC_{ANT}$ and $VOC_{BB}$, for the rpecursors of SOA. The reactivity and production of SOA from these species are discussed in further detail in Section 2.5. For bimolecular gas-phase reactions, rate constants are calculated following the Arrhenius expression

$$k = k_0 \left(\frac{T}{300}\right) exp \left(\frac{-\beta}{T}\right) \tag{1}$$

where $k_0$ is a constant, $\beta$ is the ratio of the activation energy over the universal gas constant ($E_A/R$), and T is temperature. The rate constant is then used to calculate the rate of reaction:

$$rate = k[A][B] \tag{2}$$

where $k$ is the rate coefficient, and [A] and [B] are concentrations of gases A and B, respectively.

### 2.1.1 Gaseous wet deposition

Within UKCA, wet deposition of gases is calculated as a first-order process as a function of precipitation, following Walton et al. (1988). For a detailed description of the wet deposition within UKCA, see O'Connor et al. (2014). Within each grid box, the scavenging rate, r, is calculated as follows:

$$r = S_j \times p_j(l) \tag{3}$$

where $S_j$ is the scavenging coefficient for precipitation type $j$ and $p_j(l)$ is the precipitation rate for type j from model vertical level $l$. The two precipitation types, $j$, considered are convective and large-scale. For nitric acid ($HNO_3$), the scavenging coefficient is taken from Penner et al. (1991). For all remaining species, the scavenging coefficient is calculated by scaling





down the scavenging coefficient of HNO$_3$. This is done by calculating the fraction of each species in the aqueous phase as follows:

$$f_{aq} = \frac{L \times H_{eff} \times R \times T}{1 + L \times H_{eff} \times R \times T} \tag{4}$$

where $L$ is the liquid water content, R is the universal gas constant and T is the temperature. $H_{eff}$ is the effective Henry's

coefficient, which depends on the solubility of a species and the effects of dissociation and complex formation. The effective Henry's coefficient is calculated as follows:

$$H_{eff} = k(298)exp\left(-\frac{\Delta H}{R}\left[\left(\frac{1}{T}\right) - \left(\frac{1}{298}\right)\right]\right) \times \left(1 + \frac{k_{aq}}{[H^+]}\right) \tag{5}$$

where $\Delta H$ is the enthalpy of vaporisation and $k(298)$ is the rate coefficient at 298 K. $[H^+]$ is the hydrogen ion concentration (i.e pH). All cloud liquid water droplets are assumed to have a pH of 5.0 (Giannakopoulous, 1998). $k_{aq}$ is calculated for

species which dissociuate upon dissolution, and is calvculated as follows

$$k_{aq} = k_d(298)exp\left(-\frac{\Delta H_d}{R}\left[\left(\frac{1}{T}\right) - \left(\frac{1}{298}\right)\right]\right)$$

where $k_d$ and $\Delta H_d$ are the rate coefficients anfd enthalpy of vapourisation for dissociation.

**2.1.2 Gaseous dry deposition**

Dry deposition refers to the transfer of chemical species from the atmosphere to the surface in the absence of precipitation. Dry deposition of gas-phase species within UKCA has also been described in detail before (O'Connor et al., 2014) so is only described briefly here. The dry deposition velocity ($v_d$) is calculated using a resistance-based approach (Wesely, 1989). This approach is analogous to an electrical circuit, where the transport of chemical species is dependent on three resistances, $r_a$, $r_b$,

and $r_c$:

$$v_d = \frac{1}{r_a + r_b + r_c} \tag{6}$$

The aerodynamic resistance term, $r_a$, represents the resistance to transport of chemical species through the boundary layer to a thin layer of air just above the surface. This term is calculated from the wind profile, taking into account the atmospheric stability and the surface roughness:

$$r_a = \frac{ln(z/z_0) - \Psi}{k \times u^*} \tag{7}$$

where z is the height, $z_0$ is the roughness length, $\Psi$ is the Businger dimensionless stability function, k is Karman's constant, and $u^*$ is the friction velocity.

The quasi-laminar resistance term, $r_b$, refers to the resistance to transport though the thin layer of air close to the surface. The surface resistance term, $r_c$, otherwise known as the canopy resistance term, refers to resistance to uptake at the

surface. This term is dependent on the absorbing surface as well as the physical and chemical properties of species. The canopy



resistance term is related to surface conditions, time of day, and season. There are 9 surface types considered by the model. These are broad-leaved trees, needle-leaf trees, C3 and C4 grasses, shrubs, urban, water, bare soil, and land ice. These surface types are prescribed from the International Geosphere-Biosphere Programme (IGBP) dataset (Loveland et al., 2000). Within each grid box, the multiple resistances are calculated for each surface type, and then combined to provide a grid box mean

deposition velocity and first-order loss rate.

## 2.2 Aerosol (GOMAP-mode)

The aerosol component of UKCA is the 2-moment modal version of the Global Model of Aerosol Processes (GLOMAP-mode) (Mann et al., 2010). Both aerosol mass and number are transported in seven internally mixed log-normal modes (four soluble and three insoluble). Aerosol components considered are sulphate ($SO_4$), sea salt (SS), black carbon (BC), primary organic

aerosol (POA) and secondary organic aerosol (SOA). Aerosol growth occurs via nucleation, coagulation, condensation, ageing, hygroscopic growth and cloud processing. Condensation ageing refers to the coating of hydrophilic particles, resulting in transfer to the hydrophilic mode. Here, 10 monolayers of soluble particles are assumed sufficient for condensation ageing. Dry deposition and gravitational settling of aerosol follows Slinn (1982) and Zhang et al. (2012), respectively. Grid-scale wet deposition of aerosol occurs via nucleation scavenging and impact scavenging. Subgrid-scale wet removal occurs via plume

scavenging (Kipling et al., 2013). New particle formation from binary homogenous nucleation of sulphuric acid ($H_2SO_4$) follows that described by Kulmala et al. (2006). Gaseous sulphur compounds (sulphur dioxide, $SO_2$ and dimethyl sulphide, DMS) and VOCs are oxidised, forming low volatility gases, which condense irreversibly onto pre-existing aerosol. Condensation is calculated following Fuchs (1971) which is described in Mann et al. (2010). Mineral dust is also included in the model simulations, but treated in a separate aerosol module (Woodward, 2001).

## 2.4 Emissions

The emissions used in this study are all monthly-varying decadal-average, centred on the year 2000. Anthropogenic and biomass burning gas-phase emissions are prescribed following Lamarque et al. (2010). Biogenic emissions of isoprene, monoterpene and methanol (CH3OH) are also prescribed, taken from the Global Emissions Inventory Activity (GEIA), based on Guenther et al. (1995). A diurnal cycle in isoprene emissions is imposed based on solar zenith angle. POA and BC emissions

from fossil fuel combustion are prescribed following Lamarque et al. (2010). POA and BC emissions from savannah burning and forest fires are prescribed, taken from the Global Fire Emissions Database (GFEDv2; van der Werf et al. (2010)). All carbonaceous primary emissions are emitted into the insoluble mode and transferred into the insoluble. For VOCBB, CO emissions from biomass burning were used to define its spatial distribution (Lamarque et al., 2010) and scaled to reproduce the global annual VOC total emissions from biomass burning estimated from the Emissions Database for Atmospheric

Research (EDGAR) (49 Tg (VOCBB) a-1). For VOCANT, anthropogenic emissions of benzene, toluene and xylene, were taken from Lamarque et al. (2010), and scaled to reproduce the global annual anthropogenic VOC total emissions estimated by EDGAR (127 Tg (VOCANT) a-1). Scaling both VOCBB and VOCANT to the emissions type's totals (i.e. biomass burning



and anthropogenic, respectively) represents an upper limit for the SOA precursor emissions. The emissions of VOCBB and VOCANT described here have been used in Kelly et al. 2018, with the corresponding impacts on SOA rigorously evaluated against observations. Briefly, the locations of SOA observations are well suited to constrain the anthropogenic source of SOA, but not so well suited for  evaluating VOCBB. Inclusion of VOCANT in SOA production gives rise to a substantial

improvement in model agreement with observations (Kelly et al. 2018).

## 2.5 Default Treatment of SOA

In this section, the current treatment of SOA in the UKCA model is first described, followed by descriptions of new treatments of precursor deposition and oxidation mechanisms. Within the model, SOA is treated by a coupling between the UKCA gas-

phase chemistry and GLOMAP-mode. Emitted parent hydrocarbon gases undergo a single-step oxidation, forming a secondary organic gas (SOG) which condenses, forming SOA. This is shown in Eq (8):

$$VOC + [o] \xrightarrow{k_{VOC+[O]}} \alpha_{VOC+[O]} SOG \rightarrow SOA \qquad (8)$$

where VOC is the concentration of the emitted parent hydrocarbon, [o] is the oxidant concentration, $k_{VOC+[O]}$ is the temperature-dependent rate coefficient (Eq (1)), $\alpha_{VOC+[O]}$ is the stoichiometric coefficient, and SOG is the secondary organic gas. SOG is treated as non-volatile, and although there is evidence that OA is both semi-volatile (Robinson et al., 2007;Donahue et al., 2012) and non-volatile (Jimenez et al., 2009;Cappa and Jimenez, 2010), SOG condenses irreversibly to form SOA in UKCA. The yield is identical for all oxidation reactions (13 %), regardless of VOC or oxidant. Essentially, the volatility distribution

is assumed to be identical for all reactions, irrespective of parent VOC and oxidant. In the model, no SOA precursor undergoes dry or wet deposition.

In this study, SOA production is only considered from VOCs. These include monoterpene, isoprene, VOCBB and VOCANT. Monoterpene and isoprene contain both single and double carbon bonding and therefore react with ozone ($O_3$) and the hydroxyl (OH) and nitrate ($NO_3$) radicals, forming SOG and subsequently SOA (Eq 8). Note, for isoprene, oxidation in the

context of SOA production (Eq 8) occurs independently to isoprene oxidation in the Mainz Isoprene Mechanism described in Section 2.1. Reaction kinetics for isoprene and monoterpene ($\alpha$-pinene) oxidation are taken from Atkinson and Arey (2003), and are shown in Table 1. As discussed in Section 2.4, VOCANT and VOCBB are surrogate compounds, which do not retain molecular information, and therefore, do not have laboratory derived rate constants. Initially, the assumption is made that VOCANT and VOCBB are reduced compounds, with only single carbon bonding and react predominantly with OH. VOCANT

and VOCBB are also assumed to have a similar reactivity to monoterpene towards OH oxidation, but do not react with $O_3$ or $NO_3$. These assumptions in the parent hydrocarbon reactivity are discussed further in Section 2.4.2. As stated above, none of the SOA precursors in this scheme are wet or dry deposited. In summary, the current SOA scheme suffers from a lack of





mechanistic detail in oxidation mechanisms, and neglects precursor deposition. In the following sub-sections, modifications to the model are described and the impacts of these processes quantified.

### 2.4.1 Addition of SOA Precursor Deposition

Precursors of SOA include the emitted parent hydrocarbons (monoterpene, isoprene, $VOC_{ANT}$, $VOC_{BB}$) and the secondary organic product (SOG). Several modifications were made to UKCA to investigate the influence of precursor deposition on SOA. Firstly, wet deposition of the gas-phase species, as described in Section 2.1.1, was extended to include all SOA precursors. The effective Henry's Law coefficient, for all SOA precursors, was either set to $10^5$ or $10^9$ M atm$^{-1}$. These values of $H_{eff}$ were taken from estimates by Hodzic et al. (2014). Secondly, the treatment of dry removal of gas-phase species (section 2.1.2) was extended to include all SOA precursors and they were assumed to have identical surface resistances. Table 2 shows the surface resistances for the SOA precursors over the 9 surface types. The aerodynamic and quasi-laminar surface resistances were calculated online, based on relative molecular mass and meteorology. During field studies over forested regions, organic hydroperoxides (ROOH) were observed to undergo significant dry deposition (Hall et al., 1999;Valverde-Canossa et al., 2006;Nguyen et al., 2015). Surface resistances derived from these field studies range from $5 - 40$ sm$^{-1}$ (Hall et al., 1999;Nguyen et al., 2015). Hence, these field-derived surface resistances of ROOH ('Low'; Table 2) were used to provide a lower estimate of the surface resistances of SOA precursors. Surface resistances corresponding to the dry deposition of CO ('High'; Table 2) were used to provide an upper limit of the surface resistances of SOA precursors.

### 2.4.2 Addition of a New Oxidation Mechanism for $VOC_{ANT/BB}$

As discussed in Section 2.4, initially $VOC_{ANT/BB}$ follows a single-step oxidation mechanism, with a single fixed SOA yield, and with a reactivity based on α-pinene (Table 1). However, of the anthropogenic and biomass burning VOCs related to SOA production, aromatic compounds have been identified as important components in field studies (Von Schneidemesser et al., 2010;Ding et al., 2012;Guo et al., 2012;Peng et al., 2013). Furthermore, environmental chamber studies suggest aromatic hydrocarbons undergo multi-generational oxidation reactions, with SOA yields dependent on oxidant concentrations (Ng et al., 2007;Chan et al., 2009;Kautzman et al., 2010;Li et al., 2016;Al-Naiema and Stone, 2017;Li et al., 2017b;Schwantes et al., 2017). Therefore, in order to examine how SOA is affected by variations in oxidation mechanisms, chamber-derived aromatic oxidation pathways are applied to $VOC_{ANT/BB}$. This section outlines how the chamber-derived aromatic oxidation pathway, postulated by Ng et al. (2007), is applied to the mechanistic description of SOA production from $VOC_{ANT/BB}$ within UKCA.

Figure 1 shows a mechanistic description of SOA production from toluene, accounting for the influence of NOx on SOA production, adapted from Ng et al. (2007). Briefly, toluene undergoes oxidation by OH, followed by addition of oxygen and isomerisation, to form a bicyclic peroxy radical, $RO_2$. The bicyclic peroxy radical undergoes competitive reactions with hydroperoxyl radical ($HO_2$) and NO. The $HO_2$ pathway forms non-volatile products, whereas products of the NO pathway are



semi-volatile. Although Figure 1 shows a mechanistic description of toluene oxidation, the oxidation of other methylated aromatic compounds will also follow a similar pathway. This mechanism for aromatic oxidation, as shown in Figure 1, was applied to $VOC_{ANT/BB}$ oxidation. The rate determining step in Figure 1 is the initial oxidation by OH and, therefore, the mechanism can be simplified as follows:

$$VOC + [o] \xrightarrow{k_{VOC+OH}} RO_2 \xrightarrow{k_{RO_2+HO_2}} \alpha_{RO_2+HO_2} SOG \rightarrow SOA$$
$$\xrightarrow{k_{RO_2+NO}} \alpha_{RO_2+NO} SOG \rightarrow SOA \qquad\qquad (9)$$

where VOC represents $VOC_{ANT/BB}$, $k_{VOC+OH}$ represents the rate constant for aromatic oxidation by OH, $RO_2$ represents the bicyclic peroxy radical, $k_{RO_2+HO_2}$ and $\alpha_{RO_2+HO_2}$ represent the rate constant and the stoichiometric coefficient for the $RO_2+HO_2$ reaction, respectively, and $k_{RO_2+NO}$ and $\alpha_{RO_2+NO}$ represent the rate constant and the stoichiometric coefficient for the $RO_2+NO$ reaction, respectively. Within the model, the difference in volatility distribution between the products of the $RO_2$ reactions are controlled by the stoichiometric coefficients ($\alpha_{RO_2+HO_2}$ and $\alpha_{RO_2+NO}$). Previous modelling studies use a similar method to treat

SOA production via the $RO_2+HO_2$ pathway. Assuming that the products from oxidation of explicit aromatic compounds are non-volatile, Henze et al. (2008) uses a stoichiometric yield of around 18 %. Using IVOC emissions based on naphthalene, Pye and Seinfeld (2010) uses a stoichiometric coefficient of 73 %. However, both Henze et al. (2008) and Pye and Seinfeld (2010) treat products from the $RO_2+NO$ pathway as semi-volatile, with stoichiometric yields ranging from 2 to 107 %, and equilibrium partitioning coefficients ranging from 0.0037 to 3.3150 $m^3\,\mu g^{-1}$. The reaction kinetics for aromatic oxidation used

here are shown in Table 1.

## 2.5 Model simulations

In this study, 10 simulations were performed to explore the influence of hydrocarbon deposition and oxidation mechanisms on SOA, and are described in Table 3. The duration of all simulations is two years, spanning from 1999 to 2000. The first year

was discarded as spin-up, and analysis was performed on the second year - 2000. Firstly, a control simulation was conducted, where the oxidation of all parent hydrocarbons (isoprene, monoterpene, $VOC_{ANT}$ and $VOC_{BB}$) followed Eq (8) and no SOA precursors were lost by wet or dry deposition processes. Next, the influence of VOC deposition on SOA was explored. To begin with, precursors were assumed to have low surface resistances (Low; Table 2), thus, testing the upper limit for precursor dry deposition (Dry_High; Table 3). Next, the strength of precursor surface resistance was increased (High; Table), testing the

lower limit for deposition rates (Dry_Low; Table 3). Next, SOA precursors were treated as soluble and were, therefore, included in the wet deposition scheme. As with dry removal, the upper and lower limits of precursor wet deposition were tested by carrying out two simulations, one with a higher solubility (Wet_High), and one with a lower solubility (Wet_Low). An



additional simulation was conducted to test whether the effects of precursor dry and wet deposition on SOA are additive (DryH_WetL). Note, for this simulation, dry and wet deposition are included with low surface resistances (Low; Table 2) and low solubility. Alternative combinations of surface resistances and solubility could have been used to quantify the combined influence of precursor dry and wet deposition on SOA.

Next, the influence of VOC oxidation mechanisms on SOA was explored by modifying the mechanistic description of SOA production from anthropogenic and biomass burning VOCs. As discussed in Section 1, oxidation mechanisms within SOA schemes vary substantially. Therefore, in this section, where necessary, changes to $VOC_{ANT/BB}$ oxidation were made in a step-wise fashion, in order to isolate the effects of individual changes. Firstly, the combined effects of the use of a reactive aromatic compound (naphthalene) and introducing a reaction intermediate ($RO_2$) were explored in the Multi_nap simulation,

where $VOC_{ANT/BB}$ follows Eq (9). In this simulation, stoichiometric reaction yields of 13 % are applied to both $RO_2$ oxidative pathways, which is identical to the reaction yield applied to simulations following the single-step mechanism (Eq (8)). The effects of changes to parent $VOC_{ANT/BB}$ reactivity, the chemical fate of the new reaction intermediate, and SOA production from this intermediate are discussed separately, in Sections 5.1.1, 5.1.2, and 5.1.3 respectively. Next, the influence of accounting for the difference in volatility distribution of products between the peroxy radical pathways was accounted for in

a further model experiment (Multi_nap_yield), which is discussed in Section 5.1.4. This was achieved by increasing the SOA yield from 13 to 66 % for the $HO_2$ pathway, whilst leaving the reaction yield for the NO pathway unchanged at 13 %. A stoichiometric yield of 66 % was selected as this allows quantification of the theoretical upper limit of SOA production from this pathway. $RO_2$ and SOG have differing relative molecular masses. Consequently, a stoichiometric yield of 66 % corresponds to a mass yield of 100 %. Therefore, 66 % is the highest stoichiometric yield that ensures conservation of mass

without the addition of other atoms, such as oxygen. Next, the influence of parent hydrocarbon reactivity was explored, whilst maintaining identical reaction mechanisms and yields (Section 5.1.5). In this simulation, $VOC_{ANT/BB}$ adopts the reactivity of toluene (Multi_tol_yield) and benzene (Multi_benz_yield) (Table 3). Note, for the simulations investigating the influence of oxidation mechanisms on SOA, isoprene and monoterpene oxidation is unchanged. The emissions of all SOA precursors (isoprene, monoterpene, and $VOC_{ANT/BB}$) are identical in all the simulations.

## 3 Observations used to evaluate modelled OA

This section describes the observations used to test the effects of variations in hydrocarbon physicochemical processes on model performance. To make direct comparisons, and provide a consistent method for evaluating model performance, a suite

of observations were chosen which are identical to those used in previous studies involving the UKCA model (Kelly et al., 2018).



The Aerosol Mass Spectrometer (AMS) allows on-line detection of submicron non-refractory aerosol (Jayne et al., 2000;Canagaratna et al., 2007). This method was used to measure OA concentrations for all observations utilised in this study. Uncertainties associated with this method are estimated to be between 30 and 50 % (Bahreini et al., 2009). All observations used in this study can be accessed on the AMS global network website (https://sites.google.com/site/amsglobaldatabase/).

Surface OA observations from the AMS network, originally compiled by Zhang et al. (2007), span the time period 2000-2010. The 37 observed surface measurement locations are shown in Figure 2 and coloured according to the environment sampled: urban, urban downwind, or remote. With the exception of Manaus (Brazil) (Martin et al., 2010), and Welgegund (South Africa) (Tiitta et al., 2014), all surface OA spectra were analysed further using factor analysis, classifying OA as either oxygenated OA (OOA) or hydrocarbon-like OA (HOA). Here, measured OOA is assumed comparable to modelled SOA, and

measured HOA is assumed comparable to POA. For each observation, the corresponding model-grid box was selected. Also, observations were compared to the simulated monthly-mean from the year 2000.

Observed OA concentrations from several aircraft campaigns were also used. Observation data from these aircraft campaigns, which were originally compiled by Heald et al. (2011) can also be accessed on the AMS global network website (https://sites.google.com/site/amsglobaldatabase/). Aircraft observations utilised in this study are also shown in Figure 2. These

campaigns span the period 2000 - 2010. Four campaigns were carried out in remote regions, located over the north Atlantic Ocean (TROMPEX and ITOP), Borneo (OP3) and the tropical Pacific Ocean (VOCALS-UK). Three campaigns were also carried out in polluted regions of Europe (EUCAARI, ADIENT and ADRIEX). Three campaigns were carried out in North America and were influenced heavily by biomass burning (ARCTAS-A, ARCTAS-B and ARCTAS-CARB). This observational dataset was supplemented with a campaign conducted over West Africa (AMMA;  (Capes et al., 2008)).

Observed OA from each aircraft campaign was first interpolated onto the vertical grid of the model. The model's horizontal grid cell was then matched to the observations. Again, the month of the observations were matched to the monthly mean estimate for the year 2000 simulated by UKCA. For evaluations of surface and aircraft data against simulated OA concentrations, the mismatch in measurement and simulation years is a potential contributor to the model-observation bias. This mismatch in time may be particularly important for regions influenced by biomass burining as the interannual variability

of this emissions source is substantially high  (Tsimpidi et al., 2016).

## 4 Influence of precursor deposition on SOA

In this section, the influence of VOC deposition (section 2.4.1) on simulated SOA is quantified. Next, the influence of VOC deposition on model agreement with observations is evaluated.

**4.1 Simulated SOA budget and concentrations**

When precursor deposition is neglected from the model, the simulated global annual-total SOA production rate is 75 Tg (SOA) a$^{-1}$ (Control; Table 3). The inclusion of VOC dry deposition with high surface resistances (High; Table 2) reduces the global



annual-total SOA production rate by only 2 Tg (SOA) a$^{-1}$ (2 %) (Dry_Low; Table 3). However, the rate of VOC dry deposition is highly sensitive to the value of surface resistance. The inclusion of VOC dry deposition with lower surface resistances (Low; Table 2) reduces the global annual-total SOA production rate by 24 Tg (SOA) a$^{-1}$ (32 %) (Dry_High; Table 3). Therefore, inclusion of precursor dry deposition reduces the global annual-total SOA production rate by 2-24 Tg (SOA) a$^{-1}$, or 2-32 %, with this range reflecting uncertainties in surface resistances (Table 3).

Wet removal also has a substantial impact on SOA. For example, under the assumption of an effective Henry's coefficient of $10^5$ M atm$^{-1}$, wet deposition reduces the global annual-total SOA production rate by 12 Tg (SOA) a$^{-1}$ (15 %) compared to when no precursors undergo deposition (Wet_Low; Table 3). However, as discussed in Section 1, H$_{eff}$ has been calculated to range from $10^5$ to $10^9$ M atm$^{-1}$ for VOC precursors of SOA from different sources (Hodzic et al., 2014). In this study, when H$_{eff}$ of SOA precursors is increased to $10^9$ M atm$^{-1}$, wet removal reduces the global annual-total SOA production rate by only 13 Tg (SOA) a$^{-1}$ (17 %) (Wet_High; Table 3). Therefore, the influence of precursor wet deposition on SOA is rather insensitive to uncertainties in the range of effective Henry's coefficients.

Generally, global (Hodzic et al., 2016) and regional (Bessagnet et al., 2010;Knote et al., 2015) scale modelling studies suggest that dry deposition of precursor dominates over wet deposition. Therefore, for subsequent simulations, where both dry and wet removal were included in the model (DryH_WetL), surface resistances corresponding to Dry_High, which had the largest impact on global SOA production, were used, along with H$_{eff}$ of $10^5$ M atm$^{-1}$ (Wet_Low). The influence of dry and wet deposition of precursors on the global SOA budget are not additive. The combination of dry and wet deposition of VOCs reduces the global annual-total SOA production rate by 28 Tg (SOA) a$^{-1}$ (37 %) (DryH_WetL; Table 3). Overall, deposition of SOA precursors has a substantial impact on the global SOA budget, with the global annual-total SOA production rate from all VOC source ranging from 47 to 74 Tg (SOA) a$^{-1}$, with the range reflecting uncertainties in precursor deposition (Table 3).

Figure 3 shows the sensitivity of annual-average surface SOA concentrations to precursor deposition. The spatial distribution of SOA closely reflects the location of biogenic, anthropogenic and biomass burning emissions, as noted previously (Kelly et al. 2018). Over India, extremely high anthropogenic emissions combine with moderate biogenic emissions to result in annual-average surface SOA concentrations reaching up to 17 µg (SOA) m$^{-3}$ (Figure 3 a). Over tropical forest regions of South America and Africa, biogenic and biomass burning emissions are extremely high, resulting in annual-average surface SOA concentrations ranging from 2 to 10 µg (SOA) m$^{-3}$ (Figure 3 a). Over Europe and North America, moderate emissions from anthropogenic and biogenic sources generate annual-average surface SOA concentrations in the range of 0.3 – 6 µg (SOA) m$^{-3}$ (Figure 3 a).

Over India and tropical forest regions of South America and Africa, including VOC dry deposition reduces annual-average surface SOA concentrations by 1.5 to 5 µg (SOA) m$^{-3}$ (Figure 3 e), corresponding to reductions of 15 to 50 % (Figure 3 f). Over these same regions, inclusion of precursor wet deposition reduces annual-average surface SOA concentrations by 0.5 to 1.5 µg (SOA) m$^{-3}$ (Figure 3 g), corresponding to reductions of 5 – 10 % (Figure 3 h). Over North America, annual-average surface SOA concentrations are reduced by 0.3 – 1.5 µg (SOA) m$^{-3}$ when precursor dry removal is included (Figure 3 e), corresponding to a reduction of around 20 to 35 % (Figure 3 f). Over Europe, dry deposition lowers annual-average surface



SOA concentrations by around 0.2 µg (SOA) m$^{-3}$ (25 – 40 %, Figures 3 e, f). Over both North America and Europe, the inclusion of wet deposition reduces annual average surface SOA concentrations by less than 0.2 µg (SOA) m$^{-3}$ (Figure 3 g), but this corresponds to reductions of 20 to 35 % (Figure 3 h).

Until now, the impacts of precursor deposition on SOA concentrations have only been quantified over Europe (Bessagnet et al., 2010) and North America (Knote et al., 2015). The sensitivity of SOA to precursor dry removal is in broad agreement with Bessagnet et al. (2010), who estimates that precursor dry deposition reduces July-average surface SOA concentrations by 20 – 40 % over Europe, compared to 25 - 35 % for the same period in our study. Also, Knote et al. (2015) estimates that precursor dry deposition reduces annual-average surface SOA concentrations by 46 % over North America, compared to up to 20 - 35 % in our study. The modelled sensitivity of SOA concentrations to wet deposition in this study is in relatively good agreement with Knote et al. (2015), who estimates a 10 % reduction in annual-average surface SOA concentrations over North America when precursor wet deposition is included, which agrees with the  5 - 15 % reduction found here.

When dry and wet removal of VOC precursors are both included, SOA concentrations are substantially lower. However, as noted before, the effects of these removal processes do not add linearly. Inclusion of both dry and wet deposition of SOA precursors reduces annual-average surface SOA concentrations by 25 – 40 % over most continental regions (Figure 3 d), with maximum reductions of 5 µg (SOA) m$^{-3}$ over India (Figure 3 c).

The lifetime of SOA precursors with respect to both oxidation and deposition is small. Hence, SOA precursors undergo very little transport before removal. Therefore, dry and wet deposition rates of VOCs are largest over terrestrial environments, where they are released.

## 4.2 Comparison of simulated and observed OA concentrations

In this section, the influence of SOA precursor deposition on model agreement with observations is quantified. First, simulated SOA and OA concentrations are evaluated against surface observations in the northern hemisphere (NH) and southern hemisphere (SH), respectively. Next, vertical profiles of simulated OA concentrations are compared against aircraft observations.

Figure 4 shows SOA concentrations for the simulations described in Table 2, compared to observed surface SOA concentrations across the NH mid-latitudes, which are shown in Figure 2. When deposition of SOA precursors is omitted from the model, simulated SOA concentrations are substantially lower than observed, with a normalised mean bias (NMB) of -50 % (Figure 4 a). The model negative bias is present for each site-environment type but most evident in urban environments. For several sites in urban environments, observed SOA concentrations exceed simulated SOA concentrations by greater than a factor of 10 (Figure 4 a – red triangles). The model negative bias is also consistent regionally. Without SOA precursor deposition, the NMB for Europe, North America and Asia is -50, -37 and -62 %, restively.




The model negative bias with respect to observed SOA concentrations is common among global models (Tsigaridis et al., 2014). For several modelling studies, the negative bias is primarily attributed to either underestimated reaction yields, underestimated emissions, and/or missing emissions sources. Hodzic et al. (2016) partially attributes the model negative bias with respect to observations to laboratory-derived SOA yields which do not account for wall losses. Other studies highlight

VOC emission uncertainties such as underestimates in inventories (Li et al. (2017a), or the absence of semi- and intermediate-volatility organic compounds (S/IVOCs) which can contribute to SOA (Pye and Seinfeld, 2010).

Inclusion of precursor deposition further reduces model agreement with observations. As discussed in Section 4.1, including VOC dry deposition reduces the global annual-total SOA production rate by 32 % (24 Tg (SOA) a$^{-1}$), whereas including VOC wet deposition  reduces SOA production by 15 % (12 Tg (SOA) a$^{-1}$) (Table 3). Therefore, the model negative

bias is larger when including dry deposition (NMB = -64 %; Figure 4 b) compared to that when including wet deposition (NMB = -54 %; Figure 11 c). However, as the effects of VOC precursor dry and wet removal on simulated SOA are not additive, model performance is not substantially worse when both wet and dry deposition are considered (NMB = -66 %; Figure 4 d). When the measurement sites are categorised by region, with both dry and wet removal included, the NMB across Europe, North America and Asia is -66, -53 and -77 %, respectively.

Observed and simulated OA are shown in Figure 5 for two sites in the tropics and SH, over Manaus (Brazil) and Welgegund (South Africa).  Without precursor deposition, simulated SOA is overestimated compared to observed OA over Manaus (Brazil) (Figure 5 a), but underestimated over Welgegund (South Africa) (Figure 5 b). Therefore, inclusion of precursor deposition improves model performance over Manaus (Brazil) (Figure 5 a), but not over Welgegund (South Africa) (Figure 5 b). However, the scarcity of observations in the tropics and the SH result in difficulty in drawing robust conclusions

on the influence of precursor deposition on model agreement with observations in this region.

Figure 6 shows the simulated OA vertical profiles against the AMS aircraft measurements.  Without precursor deposition, model negative biases are again evident and are largest in polluted and biomass burning influenced regions in the NH. For example, over Europe (AIDENT, ADRIEX and EUCAARI) and North America (ARCTAS-A, ARCTAS-B and ARCTAS-CARB), OA concentrations are underestimated by 71% (ARCTAS-CARB; Figure 6 j) to 97 % (ARCTAS-B; Figure

6 h) when considering all altitudes. When VOC precursors of SOA do not undergo deposition, over Western Afrcia, simulated OA concentrations are in good agreement between 0 and 3 km (Figure 6 k). However, above 3 km, model and simulated OA concentrations begin to deviate, with observed OA increasing with altitude, but modelled OA decreasing with altitude (Figure 6 k). When considering all altitudes of the AMMA campaign, modelled and measured OA concentrations are in fairly good agreement, with a NMB of -53 % (Figure 6 k).

Over North Amercia and Europe, including precursor deposition slightly worses the model negative bias. When both precursor dry and wet deposition are included, the model underestimates observed OA concentrations by 75% (ARCTAS-CARB; Figure 6 j) to 98 % (ARCTAS-B; Figure 6 h). Over West Afrcia, when VOC precursors of SOA undergo deposition, the model underestimates observed OA concentrations by 61 % (Figure 6 k).



Compared to other environments, in remote regions, model agreement with observations is relatively good, and the inclusion of precursor deposition results in both improvements and degradations in model biases in simulated OA compared to observations. Without SOA precursor deposition, simulated OA levels in VOCALS and ITOP-UK, similar to the pollution and biomass burning influenced regions, are much lower compared to observed OA (NMB = -22 and -78 %; Figure 6 a and g, respectively). Therefore, inclusion of precursor deposition further reduces model agreement with observations (NMB = -49 and -91 %; Figure 6 a and g, respectively). In contrast, for the TROMPEX and OP3 campaigns, when precursor deposition is neglected, simulated OA is higher than observed (NMB = 25 and 5 %; Figure 6 b and c, respectively). Inclusion of precursor deposition at these locations changes the model positive bias into a negative bias (NMB = -23 and -22 %; Figure 6 b and c, respectively). For all aircraft campaigns conducted in remote environments, generally, simulated OA lies within one standard deviation of the observed concentration, irrespective of whether deposition of precursors is considered or not.

Overall, the inclusion of precursor deposition influences model agreement with observations somewhat. In particular, inclusion of precursor deposition worsens model negative biases with respect to observations in the NH mid-latitudes. However, differences between simulated OA concentrations from these simulations is substantially less than the difference between simulated and observed OA. These results highlight that variations in VOC deposition contribute to considerable uncertainty in both the global SOA budget and have some impact on model agreement with observations.

## 5 Influence of aromatic oxidation mechanisms on SOA

In this section, the sensitivity of SOA to hydrocarbon oxidation mechanisms is quantified. Here, oxidation mechanisms for anthropogenic and biomass burning VOCs are modified as described in section 2.4.2. To begin with, the influence of anthropogenic and biomass burning VOC oxidation mechanisms on simulated SOA is explored. Next, the impact on model agreement with observations is evaluated. In all simulations, deposition of SOA precursors is included (Table 3), emissions of all SOA precursors are held constant, and the mechanistic description describing the oxidation of biogenic SOA precursors (monoterpene and isoprene) is held fixed, following Eq (8).

### 5.1 Simulated SOA budget and concentrations

Firstly, the single-step oxidation mechanism of $VOC_{ANT/BB}$ with reactivity based on α-pinene and a fixed reaction yield of 13 % is described (DryH_WetL). The global annual-total reaction fluxes and SOA production rates from anthropogenic and biomass burning hydrocarbons are shown in Figure 7. As described in Section 2.3, the global annual-total $VOC_{ANT/BB}$ emission rate is 176 ($VOC_{ANT/BB}$) a$^{-1}$, which is held fixed across all simulations. In this case, the global annual-total $VOC_{ANT/BB}$ oxidation rate by OH is 94 Tg (VOC) a$^{-1}$ (DryH_WetL; Figure 7). The remaining 82 Tg (VOC) a$^{-1}$ undergoes deposition (not shown). For this single-step mechanism, oxidation of the emitted parent hydrocarbon directly forms the non-volatile product, SOG,




which condenses almost immediately. A fixed reaction yield of 13 % is assumed, resulting in a global annual-total SOA production rate of 18.4 Tg (SOA) a$^{-1}$ (Figure 7). Note, due to differences in relative molecular masses for VOC$_{ANT/BB}$ and SOG, the stoichiometric yield is not equivalent to the mass yield. Expressed as a fraction of emitted parent VOC (176 Tg (VOC) a$^{-1}$), the overall yield of SOA production from anthropogenic and biomass burning VOCs (18.4 Tg (SOA) a$^{-1}$) is around 10 %.

5       The combination of a single step oxidation mechanism and the assumption of a relatively reactive parent hydrocarbon results in rapid production of SOA. Figure 8 shows the spatial distributions of annual-total surface VOC$_{ANT/BB}$ emissions, annual-average surface OH concentrations, annual-total vertically integrated VOC$_{ANT/BB}$+OH oxidation rates, and the resulting SOA production rates. As expected, the spatial distributions of VOC$_{ANT/BB}$ emissions mainly reflects anthropogenic activity. Over high emissions regions, OH concentrations are also relatively high. Over India, China, Europe and North America,

annual-average OH concentrations are in the range of 32 – 130 x10$^{-3}$ ppt(v) (Figure 8 b). Therefore, for most major VOC$_{ANT/BB}$ emissions source regions, OH availability is high, resulting in rapid oxidation; reaction fluxes of VOC$_{ANT/BB}$+OH peak very close to emissions sources (c.f. Figure 8 a, c). However, uncertainty in simulated OH concentrations will be trabslated into uncertainty in SOA production. OH is the principal oxidising agent of the atmosphere. Therefore, in order to successfully model OH, many other species (e.g. methane) also need to be modelled correctly (Lelieveld et al., 2016). Due to its very short

lifetime (~seconds) and low concentrations, OH is difficult to measure (Stone et al., 2012). Therefore, the sparsity of observations results in difficulty in constraining simulated OH concentrations in a global model.

      Also, as shown in Eq (8), oxidation of the parent VOC results in immediate production of the condensing species, SOG. Hence, not only do parent VOCs undergo rapid oxidation, but the product of this reaction is in the form of condensable organic vapours. Therefore, this combination of high parent VOC reactivity with few reaction steps results in extremely

localised SOA production from anthropogenic and biomass burning emissions. This is in contrast to other global modelling studies, which predict more regionally distributed SOA production (Pye and Seinfeld, 2010;Tsimpidi et al., 2016). Differences in the geographical extent to which SOA production occurs may be attributed to precursor reactivity and the number of reaction intermediates. For example, here, the parent hydrocarbon is a VOC, with a rate constant of 52.9 x10$^{-12}$ cm$^3$ molecule$^{-1}$ s$^{-1}$ at 298 K (Table 1), forming SOA in a single-step reaction mechanism. Hence, local SOA production is simulated (Figure 8 d).

Conversely, SOA production is more regionally distributed when treated from S/IVOC multigenerational chemistry, where the parent hydrocarbon and oxidation products all react relatively slowly (Tsimpidi et al., 2016). High observed OA concentrations over remote regions (Boreddy et al., 2015;Boreddy et al., 2016) provide evidence for the slow and sustained mechanistic description of SOA production from S/IVOCs (Tsimpidi et al., 2016). High observed OA concentrations within industrialised emissions source regions (Zhang et al., 2007) support the fast mechanistic description of SOA production from

VOCs simulated here.

      To summarise, the combination of fast reactivity and a single step oxidation mechanism favours extremely localised SOA production, with parent VOCs undergoing rapid oxidation and subsequent condensation close to source.

      In the following sub-sections, SOA formation mechanisms are altered, including increases to the number of reactions steps, accounting for the influence of oxidants on SOA yields, and reducing the chemical reactivity of the parent VOC (Eq



(9); section 2.4.2). This begins with an evaluation of the multigenerational mechanism with reactivity based on naphthalene (Multi_nap), and how this mechanism compares to the single-step oxidation mechanism with reactivity based on α-pinene (DryH_WetL). For this comparison, the individual effects of reduced parent VOC reactivity, introduction of the reaction intermediate, and SOA production from the reaction intermediate, are evaluated separately in Sections 5.1.1 to 5.1.3. Next, the

effects of accounting for the difference in volatility between $RO_2$ oxidation products is evaluated (Multi_nap_yield) in Section 5.1.4. Finally, less reactive parent hydrocarbons are explored in Section 5.1.5 (Multi_tol_yield and Multi_benz_yield).

## 5.1.1 Initial OH oxidation of parent hydrocarbon

Production of SOA from anthropogenic and biomass burning hydrocarbons is modified in the following sub-sections to follow the multigenerational mechanism of Eq (9). Naphthalene, the most reactive aromatic VOC considered in this study, is first selected (section 2.4.2), with identical reaction yields applied to both $RO_2$ pathways (Multi_nap simulation; Table 3).

The initial reaction of $VOC_{ANT/BB}$ with OH is compared to that of a single oxidation reaction step (DryH_WetL; Table

3). At 298 K, the rate constants for α-pinene and naphthalene oxidation by OH are 52.9 and 23.3 $x10^{-12}$ $cm^3$ $molecule^{-1}$ $s^{-1}$, respectively (Table 1 and 3). The global annual-total $VOC_{ANT/BB}$ oxidation rate reduces by 3 Tg (VOC) $a^{-1}$ (or 3 %), from 94 Tg (VOC) $a^{-1}$ using the reactivity of α-pinene, to 91 Tg (VOC) $a^{-1}$ using the reactivity of naphthalene (Figure 7). Therefore, the global $VOC_{ANT/BB}$ oxidation rate is relatively insensitive to a ~50 % reduction in reactivity. When applying a 13 % stoichiometric yield to this reaction sequence (Table 3), this reduction in parent VOC oxidation rate contributes to a marginal

change in the global annual-total SOA production rate (0.6 Tg (SOA) $a^{-1}$).

The response of regional VOC oxidation rates to a ~50 % reduction in the reactivity vary in both magnitude and sign. Figure 9 shows the difference in annual-total vertically integrated $VOC_{ANT/BB}$ oxidation rates for all the multigenerational oxidation mechanism simulations in Table 3, relative to the single oxidation pathway with reactivity based on α-pinene (DryH_WetL; Table 3). Reduced chemical reactivity lowers oxidation rates within emission source regions. For example, over

India and parts of Africa, annual-total $VOC_{ANT/BB}$ oxidation rates reduce by up to 0.05 Tg ($VOC_{ANT/BB}$) $a^{-1}$ (Figure 9 a); these changes in annual-total $VOC_{ANT/BB}$ oxidation rates within emissions source regions correspond to reductions between 10 and 30 % (not shown). By contrast, downwind of many emissions source regions, the lower reactivity acts to enhance $VOC_{ANT/BB}$ oxidation rates. For example, over the Arabian Sea, over Southeast China, off the coast of Nigeria, and over the southeast USA, annual-total $VOC_{ANT/BB}$ oxidation rates increase by 0.001 – 0.05 Tg ($VOC_{ANT/BB}$) $a^{-1}$ in response to a ~50 % reduction

in parent VOC reactivity (Figure 9 a). These changes in annual-total $VOC_{ANT/BB}$ oxidation rates downwind of emissions source regions correspond to reductions which exceed 60 % (not shown). As discussed in Section 5.1, adoption of the reactivity of α-pinene for the $VOC_{ANT/BB}$+OH reaction results in peak VOC oxidation rates at emission source, with VOCs undergoing very



little transport (Figure 8 c). Therefore, by reducing the reactivity by ~50 %, fewer $VOC_{ANT/BB}$ are oxidised at source but transport of $VOC_{ANT/BB}$ away from source is promoted.

### 5.1.2 Chemical fate of the new reaction intermediate, $RO_2$

With a multi-generational pathway, oxidation of the parent VOC forms a new reaction intermediate, the peroxy radical $RO_2$. In this case, $VOC_{ANT/BB}$ oxidation results in a global annual-total peroxy radical production rate of 91 Tg ($RO_2$) $a^{-1}$ (Multi_nap simulation; Figure 7). Introduction of this new reaction intermediate has the potential to either reduce and/or delay SOA production, depending on assumptions regarding the strength of deposition and chemical reactivity of this intermediate. For example, SOA production would be reduced if the peroxy radical undergoes significant deposition, which is dependent on

deposition parameters such as surface resistances and solubility (section 2.4.1). Additionally, SOA production could be reduced or delayed if the chemical removal of $RO_2$ is slow. The influence of introducing the peroxy radical as a reaction intermediate is therefore predetermined by assumptions in deposition parameters and reaction kinetics. In all simulations, $RO_2$ is assumed to have identical solubility and surface resistances to all other SOA precursors, $H_{eff} = 10^5$ M atm$^{-1}$ and 'Low' surface resistances (Table 2). At 298 K, the rate constants for $RO_2$ oxidation by $HO_2$ and NO, taken from Atkinson and Arey (2003),

are 14.8 and 8.5 x$10^{-12}$ cm$^3$ molecule$^{-1}$ s$^{-1}$, respectively (Table 1). Consequently, of the 91 Tg of $RO_2$ generated annually, oxidation by NO and $HO_2$ removes 57 and 34 Tg ($RO_2$) $a^{-1}$, respectively (Multi_nap_yield; Figure 7). Deposition of $RO_2$ is inconsequential at 0.1 Tg ($RO_2$) $a^{-1}$ (not shown). This extremely low deposition rate is because the chemical removal of the peroxy radical is extremely fast. The global annual-average lifetime of $RO_2$ with respect to oxidation is ~1 day, which is relatively short in comparison to atmospheric transport timescales. Therefore, due to marginal deposition and fast oxidation,

introduction of the peroxy radical reaction intermediate will probably have no effect on either the SOA production rate or the geographical distribution of SOA production, which are both quantified in the following section (5.1.3).

Chemical removal of the peroxy radical via the two oxidative pathways is an important factor in governing the strength of SOA production, as discussed later in Sections 5.1.3 to 5.1.5. $RO_2$ is chiefly removed by NO, as opposed to $HO_2$ radicals. This is demonstrated in Figure 10, which shows the relative contributions of the $HO_2$ and NO peroxy radical oxidative

pathways to the total chemical removal of $RO_2$ (top row) and to SOA production (bottom row). On a global and annual mean basis, removal by NO accounts for 62 % of $RO_2$ chemical loss (Figure 10 a). Other global modelling studies which consider the peroxy radical as a reaction intermediate from aromatic compounds or IVOCs, also predict $RO_2$ removal to be dominated by NO. Henze et al. (2008) estimate that, for peroxy radicals generated from benzene, xylene and toluene, 61 % react via the NO pathway. Peroxy radicals generated from IVOCs, with parent hydrocarbon reactivity based on naphthalene, 66 % are

consumed by NO (Pye and Seinfeld, 2010). These results suggest that the chemical fate of the peroxy radical is robust despite the likelihood of variations in precursor emissions and oxidant concentrations between this and the aforementioned studies.

The substantial preference for $RO_2$ radicals to react via the NO pathway instead of the $HO_2$ pathway can be attributed to differences in oxidant availability (i.e. concentrations) and in reaction rates. Note, in the UKCA model, $HO_2$ is assumed to





undergo wet removal. Firstly, consider the difference in oxidant levels. Figure 11 shows the spatial distribution of annual-average surface concentrations of NO and $HO_2$, as well as the ratios $NO/HO_2$ and $(k_{RO2+NO} \times NO) / (k_{RO2+HO2} \times HO_2)$. NO is extremely spatially heterogeneous (Figure 11). Within the model, sources of NOx include the prescribed anthropogenic, biomass burning and soil emissions, as well as lightning-NOx which is calculated interactively. At the surface, the highest

annual-average surface NO concentrations (1-23 ppb(v)) are simulated over industrialised and urban regions of North America, China and Europe, as well as, over the Amazon and Congo regions (Figure 11 a). Over remote marine environments, away from anthropogenic and biomass burning sources, concentrations of NO are low (Figure 11a). In contrast, concentrations of $HO_2$ are much lower and more evenly distributed across the surface (Figure 11). Over the majority of both continental and marine regions, annual-average surface $HO_2$ concentrations range between 2 and 23 ppt(v) (Figure 11b). Therefore, over most

environments, NO concentrations are far greater, with annual-average surface NO concentrations ranging from 10 ($NO/HO_2$ = $10^1$) to 10,000 ($NO/HO_2 = 10^4$) times more than $HO_2$ (Figure 11 c). Only in the remote marine environments are $HO_2$ levels higher in absolute magnitude compared to NO, with simulated annual-average surface $HO_2$ concentrations reaching 10 times that of NO ($NO/HO_2 = 10^{-1}$; Figure 11 c). At higher levels, $NO/HO_2$ reduces, suggesting an increasing importance of the $HO_2$ pathway at higher altitudes. However, due to the fast chemical reactivity, the majority of SOA production occurs at the surface.

For the majority of the atmosphere, the difference in the magnitudes of the oxidant concentrations favours the $RO_2$+NO pathway over the $RO_2$+$HO_2$ pathway.

     Differences in reactivity of $RO_2$ with respect to the oxidants also affects the fate of this radical. At 298 K, the rate constant for $RO_2$+NO is 8.42 x $10^{-12}$ $cm^3$ $molecule^{-1}$ $s^{-1}$, almost half that of $RO_2$+$HO_2$ (k(298 K) = 14.7 x$10^{-12}$ $cm^3$ $molecule^{-1}$ $s^{-1}$; Table 1). Therefore, the higher rate constant for oxidation by $HO_2$ in comparison to NO favours the $RO_2$+$HO_2$ pathway.

The ratio, $(k_{RO2+NO} \times NO) / (k_{RO2+HO2} \times HO_2)$, combines the difference in rate constants together with differences in the ratio of oxidant concentrations, and ranges from $10^0$ to $10^4$ over most continental regions, but is as low as $10^{-2}$ over remote marine environments, such as the Pacific Ocean and South Atlantic Ocean (Figure 11 d). Hence, the net effect of differences in oxidant concentrations and rate constants is to favour peroxy radical removal via the NO oxidative pathway (Figure 10 a; Figure 11d). This preference for the NO radical pathway is enhanced even further by considering the likelihood of $RO_2$ being

co-located with NO. $RO_2$ is a second generation oxidation product of $VOC_{ANT/BB}$, which is released by anthropogenic and biomass burning sources. NO emissions are predominantly emitted from anthropogenic and biomass burning sources. Therefore, peroxy radicals are very likely to be formed in NO-rich environments, further favouring the probability of entering the $RO_2$+NO pathway. Furthermore, adoption of naphthalene reactivity for $VOC_{ANT/BB}$, which is still relatively high, prevents transport away from high-NO regions. Overall, peroxy radicals preferentially react via the NO pathway due to relatively higher

NO concentrations than $HO_2$, despite the $HO_2$ pathway having a higher rate constant.



### 5.1.3 Production of SOA from new reaction intermediate, RO₂

For this multi-step reaction scheme with parent VOC reactivity based on naphthalene (Multi_nap), the initial oxidation and subsequent reaction of the intermediate were discussed in Sections 5.1.1 and 5.1.2, respectively. In this section, the production of SOA from this mechanism is examined. In this oxidation scheme, identical reaction yields of 13 % are applied

for both the $HO_2$ and NO pathways. For the $RO_2$+NO reaction, a global annual-total reaction flux of 57 Tg $(RO_2)$ $a^{-1}$ results in an SOA production rate of 11 Tg (SOA) $a^{-1}$ (Multi-nap; Figure 7). Similarly, for the $RO_2$+$HO_2$ pathway, a global annual-total reaction flux of 34 Tg $(RO_2)$ $a^{-1}$ results in an SOA production rate of 7 Tg (SOA) $a^{-1}$ (Figure 7). Hence, the relative contribution of the $RO_2$ oxidative pathways to SOA production is simply a reflection of the relative contribution of each pathway to $RO_2$ consumption. Therefore, the $RO_2$+NO pathway accounts for 62 % of the global annual-total $RO_2$ oxidation

rate (Figure 10 a), and also accounts for 62 % of the annual-total SOA production rate from anthropogenic and biomass burning hydrocarbons (Figure 10 e). The sum of global annual-total SOA production from anthropogenic and biomass burning sources, from both oxidative pathways, is 17.8 Tg (SOA) $a^{-1}$ (Figure 7). This is just 0.6 Tg (SOA) $a^{-1}$ (or 3 %) less than the global annual-total SOA production rate when using a single-step oxidation mechanisms with reactivity based on α-pinene (DryH_WetL; Figure 7). Note, this 0.6 Tg (SOA) $a^{-1}$ reduction in SOA production is solely due to the 3 % reduction

in the $VOC_{ANT/BB}$ oxidation rate (Section 5.1.1.). This therefore confirms that, due to the marginal deposition rate of $RO_2$, the introduction of the reaction intermediate has no effect on global SOA production.

       The difference in annual-average surface SOA concentrations for the multigenerational oxidation mechanisms relative to the single step reaction with reactivity based on α-pinene are shown in Figure 12. The effects of a ~50 % reduction in parent VOC reactivity in combination with the introduction of the reaction intermediate on regional annual-average surface

SOA concentrations vary in both magnitude and sign but, generally, are small. These differences in SOA concentrations (Figure 12 a and b) closely resemble differences in parent VOC oxidation rates in response to the change in chemical reactivity (Figure 9 a). Over regions where reduced reactivity has lowered $VOC_{ANT/BB}$ oxidation rates, such as India and and industrialised parts of Africa (Figure 9 a), annual-average surface SOA concentrations have reduced by around 0.1 to 0.5 µg (SOA) $m^{-3}$ (Figure 12 a), corresponding to reductions of 5 – 20 % (Figure 12 b). On the other hand, for some downwind regions, such as

Northern India, Southeast China and Southeast USA, annual-average surface SOA concentrations increase by 0.1 – 4 µg (SOA) $m^{-3}$ (Figure 12 a), corresponding to increases of 5 – 30 % (Figure 12 b). Overall, annual-average surface SOA concentrations change by less than 3 % (not shown) and the global annual-average SOA burden changes by less than 1 % (not shown). The strong similarity between the difference in SOA concentrations (Figure 12 a) and VOC oxidation rates (Figure 9 a) also confirms how introduction of the reaction intermediate did not affect the geographical distribution of SOA production.

30        To summarise, moving from a single-step oxidation mechanism with the reactivity of α-pinene and with a single SOA yield, to a multi-step oxidation mechanism with a slower reactivity based on naphthalene with a single SOA yield has very little effect on SOA production and surface concentrations. The slower reactivity of naphthalene reduces the global



VOC$_{ANT/BB}$ oxidation by 3%, contributing to a reduction in the global annual-total SOA production rate of 0.6 Tg (SOA) a$^{-1}$ (3 %). Introduction of the reaction intermediate, but with no change to reaction yields, has no effect on global SOA.

### 5.1.4 Accounting for the difference in volatility between HO$_2$ and NO oxidation products

In this section, the effects of accounting for the difference in volatility between RO$_2$ oxidation products is examined. This is done by altering the reaction yields for RO$_2$ reactions, whilst maintaining the same chemical mechanism (Eq (9)) and precursor emission rate. As discussed in Section 1, for aromatic compounds, the volatility and, therefore, the amount of SOA produced, depends on the concentrations of NOx (Hurley et al., 2001;Song et al., 2005;Ng et al., 2007;Chan et al., 2009). One explanation for this relationship is that the HO$_2$ pathway forms non-volatile products, whereas the NO pathway forms semi-volatile products. As semi-volatile compounds have a greater propensity to be in the gas phase, this explains why observed SOA yields are higher in low-NOx conditions. Hence in a further simulation, the difference in volatility between products of different peroxy radical oxidation pathways are accounted for, whereby the yield for the RO$_2$+HO$_2$ reaction is increased from 13 to 66 %, whilst the yield for the RO$_2$+NO reaction is left at 13 % (Multi_nap_yield; Table 3). Increasing the molar yield of SOG production from the RO$_2$+HO$_2$ reaction can be considered as equivalent to assuming a greater fraction of products are non-volatile. As discussed in Section 2.5, the assumption of a 66 % stoichiometric reaction yield was selected as it corresponds to a 100 % mass yield and therefore allowing the theoretical upper limit of SOA production via the HO$_2$ pathway to be quantified whilst conserving mass. With a higher molar yield of 66 %, global SOA production from the RO$_2$+HO$_2$ reaction increases to 34 Tg (SOA) a$^{-1}$ as compared to 7 Tg (SOA) a$^{-1}$ using a 13 % yield for this reaction (Figure 7). As a consequence of this increase to the hydroperoxyl reaction yield, the HO$_2$ pathway now accounts for 75 % of SOA production from anthropogenic and biomass burning sources (Figure 10 f), despite only 38 % of the RO$_2$ radicals reacting via this pathway (Figure 10 b). This is in remarkably good agreement with previous studies. Pye and Seinfeld (2010) also estimate that the HO$_2$ pathway accounts for 75 % of SOA production from I-VOCs. In addition, Henze et al. (2008) estimates that, for SOA production from benzene, toluene and xylene, 72 % is produced via the HO$_2$ pathway.

Accounting for differences in volatility between RO$_2$ oxidation products increases the global SOA production rate by 27.3 Tg (SOA) a$^{-1}$ (or 153 %), from 17.8 Tg (SOA) a$^{-1}$ when a molar yield of 13 % is applied to both pathways (Multi_nap), to 45.1 Tg (SOA) a$^{-1}$ when a molar yield of 66 % is applied (Multi_nap_yield). Under these conditions, the overall aerosol yield from anthropogenic and biomass burning VOC emissions is 25 %, which lies within the range from other modelling studies, either based on explicit aromatic compounds or IVOCs, which range from 22 – 30 % (Henze et al., 2008;Pye and Seinfeld, 2010).

The relative spatial homogeneity of HO$_2$ radicals over land and ocean, as shown in Figure 11b, suggests that increasing the yield for this pathway could lead to enhanced SOA production globally. However, as discussed in Section 5.1.1, the naphthalene+OH rate constant results in relatively fast oxidation rates. Therefore, RO$_2$ radicals are still being generated





close to the emissions source. For these reasons, increasing the reaction yield for the $HO_2$ reaction pathway increases SOA concentrations mainly over major anthropogenic emission source regions (Figure 12 c, d). In response to this increased yield, over India, China, Africa and Europe, annual-average surface SOA concentrations have increased by $0.5 - 8$ µg (SOA) m$^{-3}$ (Figure 12 c), corresponding to increases of $10 - 100$ % (Figure 12 d). Note, differences in SOA concentrations are positive

everywhere, whereas both positive and negative changes were found when comparing differences in SOA concentrations between multi-generational and single oxidative pathways results without accounting for volatility changes (c.f. Figure 12 b and d).   In summary, both globally (Figure 7) and regionally (Figure 12 d), when accounting for the different SOA yields for the $RO_2$ oxidative pathways, despite a reduction in $VOC_{ANT/BB}$ global SOA production rates, surface SOA concentrations increase everywhere.   Therefore, the lower reactivity in $VOC_{ANT/BB}$ is compensated for by lower volatility products from the

$HO_2$ oxidation pathway leading to net increases in modelled SOA.

### 5.1.5 Production of SOA from less reactive hydrocarbons

As discussed in Section 2.4, $VOC_{ANT/BB}$ is a lumped species, and, hence, represents a mixture of species with a range of

physicochemical properties. In this section, the uncertainty related to its chemical reactivity and the effects on SOA production are explored. At 298 K, the rate constant for aromatic compounds with respect to OH oxidation ranges from 1.22 to 23.2 x10$^{-12}$ cm$^3$ molecule$^{-1}$ s$^{-1}$, respectively (Table 1 and 3). Therefore, adoption of the naphthalene reactivity in the multi-generational pathway simulations described above represents an upper limit for the $VOC_{ANT/BB}$ oxidation rate when considering SOA relevant aromatic compounds. In this section, the VOC reactivity is varied across a series of different aromatic compounds:

naphthalene, toluene and benzene (Multi_nap_yield, Multi_tol_yield and Mult_benz_yield; Table 3).   However, the mechanistic description and stoichiometric yields describing SOA formation from $VOC_{ANT/BB}$ are identical and follow Eq (9).

    Firstly, consider how reactivity affects SOA production among the multigenerational oxidation mechanisms (Multi_nap_yield, Multi_tol_yield and Multi_benze_yield). Reducing the chemical reactivity of $VOC_{ANT/BB}$ reduces oxidation, whilst at the same time, favours the likelihood of $RO_2$ radicals entering the high-yield $HO_2$ pathway. The global annual-total

$VOC_{ANT/BB}$ oxidation rates are 91, 65 and 32 Tg ($VOC_{ANT/BB}$) a$^{-1}$ using the reactivity of naphthalene, toluene and benzene, respectively (Figure 7). Hence, as reactivity is reduced, oxidation is lowered at the expense of deposition. In response to this reduced oxidation rate, fewer $RO_2$ radicals are being generated, which therefore, drives reductions in SOA production. The global annual-total SOA production rates are 45.1, 34.0, 17.9 Tg (SOA) a$^{-1}$ using the reactivity of naphthalene, toluene and benzene, respectively (Figure 7). However, as the reactivity is reduced, the chances of $RO_2$ radicals entering the high-yield

$HO_2$ pathway is increased, therefore, slightly offsetting the effects of the reduced $RO_2$ production rate. The fraction of peroxy radicals entering the $HO_2$ pathway is 38, 41 and 46 % using the reactivity of naphthalene, toluene and benzene, respectively (Figure 10 d, e and h, respectively). As shown in Figure 11 d, the $HO_2$ pathway dominates only in remote marine environments. Hence, as the reactivity of the parent hydrocarbon is reduced, $VOC_{ANT/BB}$ oxidation rates close to emissions sources reduce,



but increase further downwind (Figure 9 c and d). Therefore, lower reactivity enhances the likelihood of peroxy radicals being generated downwind of emissions sources, where the $HO_2$ pathway is favoured. These findings are consistent with Henze et al. (2008), who predicted increased fluxes through the $HO_2$ pathway for peroxy radicals derived from less reactive parent aromatic hydrocarbons. Overall, reduced parent hydrocarbon reactivity reduces the sources of peroxy radicals but favours

lower volatility $RO_2 + HO_2$ oxidation products.

Secondly, consider the net effects of using aromatic oxidation to describe SOA production from $VOC_{ANT/BB}$ (Multi_nap_yield, Multi_tol_yield and Multi_benze_yield), versus using the single-step mechanism with reactivity based on α-pinene (DryH_WetL). Compared to α-pinene, the aromatic compounds, naphthalene, toluene and benzene are 50, 75 and 95 % less reactive, respectively (Table 2). As discussed in Section 5.1.3, using the chemical reactivity of naphthalene compared

to monoterpene leads to a 3 % reduction in $VOC_{ANT/BB}$ oxidation, which drives a 0.6 Tg (SOA) a$^{-1}$ (1 %) reduction in global annual-total SOA production (c.f. DryH_WetL and Multi_nap; Figure 7). However, as shown in Section 5.1.4, this reduction in VOC oxidation is entirely offset by accounting for the high-yield pathway of the $RO_2+HO_2$ reaction, leading to a 27.3 Tg (SOA) a$^{-1}$ (153 %) increase in global annual-total SOA production (c.f. DryH_WetL and Multi_nap_yield; Figure 7). Using the chemical reactivity of toluene compared to α-pinene also reduces the $VOC_{ANT/BB}$ oxidation, but this time by 31 % (c.f.

DryH_WetL and Mutli_tol_yield; Figure 7). However, similar to the case of naphthalene, this reduction in $VOC_{ANT/BB}$ oxidation is still outweighed by accounting for the high-yield $HO_2$ pathway, such that global annual-total SOA production increases by 15.6 Tg (SOA) a$^{-1}$ (or 85 %), from 18.4 Tg (SOA) a$^{-1}$ in the single step oxidation mechanism based on α-pinene, to 34.0 Tg (SOA) a$^{-1}$ in the multi-step oxidation mechanisms based on toluene (c.f. DryH_WetL and Mutli_tol_yield; Figure 7). On the other hand, benzene is considerably less reactive than α-pinene, leading to 66 % reduction in the global annual-total

$VOC_{ANT/BB}$ oxidation rate (c.f. DryH_WetL and Mutli_benz_yield; Figure 7). In this case, the reduction in $VOC_{ANT/BB}$ oxidation is so large, that it is not compensated for by accounting for the difference in volatility between $RO_2$ oxidation products. Hence, using the reactivity of benzene, the global annual-total SOA production rate reduces by 0.5 Tg (SOA) a$^{-1}$ (or 3 %), from 18.4 Tg (SOA) a$^{-1}$ in the single step oxidation mechanism based on α-pinene, to 17.9 Tg (SOA) a$^{-1}$ in the multi-step oxidation mechanisms based on benzene (c.f. DryH_WetL and Mutli_benz_yield; Figure 7). These results demonstrate

how, from a global perspective, the combined effects of introduction of the peroxy radical intermediate which also accounts for the difference in SOA yields between $HO_2$ and NO pathways can either lead to an increase (Multi_nap_yield and Multi_tol_yield) or reduction (Multi_benze_yield) in SOA production that, critically, depends on the assumed chemical reactivity of the parent VOC.

The spatial distribution of SOA is also influenced by these changes in $VOC_{ANT/BB}$ oxidation mechanisms. For cases where reactivity is based on either naphthalene (Figure 12 c and d) or toluene (Figure 12 e and f), accounting for the high yield $HO_2$ pathway compensates for reduced reactivity, such that annual-average surface SOA concentrations increase globally in comparison to the single step oxidation mechanism with reactivity based on α-pinene (DryH_WetL). The spatial pattern for the multigenerational oxidation mechanism based on benzene (Multi_benz_yield) and the single-step oxidation mechanism



based on α-pinene (DryH_WetL) are also very different (Figure 12 g and h), despite only a small difference in the global annual-total SOA production rate (Figure 7); in the multigenerational oxidation mechanism with reactivity based on benzene, $VOC_{ANT/BB}$ has slowed down substantially, and newly introduced $RO_2$ radicals are being formed in downwind environments, leading to reduced SOA concentrations in emissions sources regions, but increased SOA concentrations downwind. Over

emissions source regions, such as China, India and North America, annual-average surface SOA concentrations are lower by up to 4 µg (SOA) m$^{-3}$ (Figure 12 g). Over continental outflow regions, such as the Arabian Sea and the Bay of Bengal, annual-average surface SOA concentrations have increased by $0.1 - 0.5$ µg (SOA) m$^{-3}$ (Figure 12 h). Although the global annual-total SOA production rates are identical, the global annual-average SOA burden is 10 % greater when using benzene as the parent VOC undergoing multi-generational oxidation, highlighting the strong spatial gradients in SOA lifetime. The spatial pattern

simulated in the multigenerational oxidation pathway with reactivity based on benzene, is in greater agreement with the more regionally distributed SOA concentrations simulated in models based on S/IVOC sources (Pye and Seinfeld, 2010;Tsimpidi et al., 2016).

## 5.2 Comparison of simulated and observed OA concentrations

In this section, the influence of anthropogenic and biomass burning hydrocarbon oxidation mechanisms on model agreement with observations is quantified. Reduced parent hydrocarbon reactivity combined with accounting for the different SOA yield pathways of the peroxy radical affects model agreement with observations. Figure 13 shows simulated versus observed surface SOA concentrations for the NH from the simulations described in Table 3. In the multi-step oxidation pathway simulations,

using naphthalene and toluene, the annual-total SOA production rate increased relative to the single step fast oxidation pathway. This increase was due to the difference in volatility between products of the peroxy radical oxidation pathways, despite the reduction in parent hydrocarbon reactivity. Therefore, simulated SOA concentrations are in closer agreement to observations (Multi_nap_yield; NMB = -46 %; Figure 13 b and Multi_tol_yield; NMB = -56 %; Figure 13 c) compared to the values using the single oxidation pathway (NMB = -66 %; Figure 4 d). However, simulated SOA concentrations have the

largest negative bias for the multi-step simulation with benzene as the parent hydrocarbon (NMB = -71 %; Figure 13 d). Global annual-total emissions of benzne and toluene are 5.6 and and 6.9 Tg (C) a$^{-1}$, resptively (Henze et al., 2008), whereas emissions of naphthalene are 0.22 Tg (C) a$^{-1}$ (Pye and Seinfeld, 2010). This suggests benzene and tolune could be more realistic surrogate compounds to represent $VOC_{ANT/BB}$ mchemistry, as opposed to naphthalene. This is due to the slow reactivity of benzene resulting in a small $VOC_{ANT/BB}$ oxidation rate, which is higher downwind of emissions compared to the point of emissions

(Figure 12 h). Figure 12 demonstrates that mechanisms of oxidation have a strong influence on model agreement with observations. However, the model negative bias is persistent in all simulations, despite the oxidation pathways spanning a wide range of both chemical reactivity and reaction yields.



For the aircraft campaigns, mechanisms of anthropogenic and biomass burning oxidation have a limited influence on model agreement with observations. For the campaigns in remote regions, VOCALS (Figure 6 a), TROMPEX (Figure 6 b) and OP3 (Figure 6 c), and over Western Afrcia (AMMA; Figure 6 k), introduction of the reaction intermediate combined with a reduction in reactivity (c.f. DryH_WetL and Multi_nap) has no effect on the NMB. However, the multi-step reaction

mechanisms which do account for the high yield pathways have a substantial impact on the NMB; with reactivity based on naphthalene or toluene, the NMB reduces (Multi_nap_yield and Multi_tol_yield), but the NMB increases when the reactivity is based on benzene (Multi_benz_yield). Contrastingly, model performance in Europe and North Amercia = (Figure 6 h - j) remains similar as $VOC_{ANT/BB}$ oxidation is modified. This warrants further discussion. As explained in previous sections, the global SOA production rate is extremely sensitive to the mechanisms of $VOC_{ANT/BB}$ oxidation. However, model performance

over the pollution and biomass burning influenced regions is relatively insensitive to VOC oxidation mechanisms. This is likely to be a reflection of the location of aircraft campaigns and how they are categorised. For example, the aircraft campaigns categorised as influenced by biomass burning are in North America, but peak biomass burning emissions are located over tropical forest regions of South America and Africa. Furthermore, the aircraft campaigns categorised as influenced by pollution are all in Europe. Again, this does not correspond to the location of peak anthropogenic emissions over Asia. Therefore,

mechanisms of anthropogenic and biomass burning oxidation have substantial impacts on simulated SOA production rates, but almost no effect on model agreement with aircraft observations in 'pollution and biomass burning influenced' regions, due to a lack of aircraft coverage.

## 6. Conclusions

In this study, the description of both deposition and oxidation for SOA precursors was developed in a global chemistry-climate model. Several model integrations were conducted and the treatments of deposition and oxidation mechanisms of SOA precursors were varied. Subsequent effects on the global SOA budget were quantified and simulated OA was evaluated against a suite of surface and aircraft campaigns spanning both the southern and northern hemispheres.

Within UKCA, SOA formation is considered from VOCs – monoterpene, isoprene, a lumped anthropogenic VOC

($VOC_{ANT}$) and a lumped biomass burning VOC ($VOC_{BB}$). Under the assumption that no precursors undergo deposition, the global annual-total SOA production rate is 75 Tg (SOA) a$^{-1}$ and simulated OA concentrations are generally lower than observed (NMB = -50 %). Extending deposition to include SOA precursors has substantial impacts on both the global SOA budget and model agreement with observations. Including SOA precursor dry deposition reduces the global annual-total SOA production rate by 2 – 24 Tg (SOA) a$^{-1}$ (2 - 32 %), with the range reflecting uncertainties in surface resistances. Including SOA precursor

wet deposition reduces the global annual-total SOA production rate by 12 Tg (SOA) a$^{-1}$ (15 %) and is relatively insensitive to changes in effective Henry's Law coefficient. The effects of dry and wet deposition on the global SOA budget are not additive; the inclusion of both these processes reduces the global annual-total SOA production rate by 28 Tg (SOA) a$^{-1}$ (37 %). Inclusion of VOC deposition generally increases model negative biases with respect to observations. For SOA, across northern





hemisphere mid-latitude sites, inclusion of both dry and wet deposition of VOCs increases the NMB from -50 to -66 %. However, for OA, over Manaus (Brazil), when precursor deposition is neglected from the model, simulated OA concentrations exceed observed OA concentrations.

Production of SOA from aromatic compounds, which are typically emitted from anthropogenic and biomass burning activities, has been partially elucidated by environmental chamber studies. Briefly, parent aromatic hydrocarbons are oxidised by the hydroxyl radical (OH) to form a reaction intermediate, the peroxy radical ($RO_2$). $RO_2$ undergoes competitive reactions; with $HO_2$ the products are non-volatile, whereas with NO the products are semi-volatile. Hence, higher $HO_2$ concentrations favour higher yields of SOA.

The influence of VOC oxidation mechanisms on the global SOA budget was also examined. For the anthropogenic and biomass burning sources of SOA ($VOC_{ANT/BB}$), a series of simulations were performed with varying a) parent hydrocarbon reactivity, b) number of reaction intermediates, and c) accounting for differences in volatility between oxidation products from various pathways. The global annual-total SOA production rate from anthropogenic and biomass burning sources is 18.4 Tg (SOA) $a^{-1}$ when the parent hydrocarbon, $VOC_{ANT/BB}$, undergoes a single-step oxidation, with a fixed reaction yield of 13 %, and a reactivity based on α-pinene. Using the reactivity of naphthalene, toluene or benzene, the global annual-total $VOC_{ANT/BB}$ oxidation rate changes by -3, -31 or -66 %, respectively, when compared to using the reactivity of α-pinene. Increasing the number of reaction intermediates, by including $RO_2$ as a product of $VOC_{ANT/BB}$ oxidation, slightly delays SOA production but has no effect on the global SOA production rate. Hence, when the reactivity of $VOC_{ANT/BB}$ is reduced from α-pinene to naphthalene, in combination with the introduction of the reaction intermediate, the global annual-total SOA production rates changes by just -0.6 Tg (SOA) $a^{-1}$ (or -3 %), from 18.4 Tg (SOA) $a^{-1}$ to is 17.8 Tg (SOA) $a^{-1}$. However, the subsequent competitive chemical reactions of $RO_2$ control the volatility distribution of products. To account for this, the reaction yield for the $RO_2+HO_2$ pathway was increased from 13 to 66 %. The reaction yield for the $RO_2+NO$ pathway was left unchanged, at 13 %. Accounting for the difference in volatility between $RO_2$ products increases the global annual-total SOA production rate from anthropogenic and biomass burning by 153 %, from 17.8 Tg (SOA) $a^{-1}$ in the simulation with yields of 13 % for both $RO_2$ reactions, to 45.1 Tg (SOA) $a^{-1}$ when the yield for the $RO_2+HO_2$ is increased 66 %.

Overall, the effects of using aromatic oxidation to describe SOA formation from anthropogenic and biomass burning compounds versus using a single-step mechanism with reactivity based on α-pinene, can be explained in terms of reductions in parent VOC reactivity and accounting for the high-yield $HO_2$ pathway, as opposed to the introduction of the reaction intermediate. For both naphthalene and toluene, reduced reactivity in comparison to α-pinene is small, and is entirely offset by accounting for the difference in volatility between $RO_2$ oxidation products. By contrast, benzene is significantly less reactive than α-pinene, and accounting for the different in volatility between $RO_2$ oxidation products cannot outweigh this. For example, for naphthalene, changes in oxidation rate (-3 %) are outweighed by accounting for the difference in volatility between $RO_2$ reactions, such that the global annual-total SOA production rate changes by 27.3 Tg (SOA) $a^{-1}$ (or 145 %), from 18.4 Tg (SOA) $a^{-1}$ in the single step oxidation mechanism based on α-pinene to 45.1 Tg (SOA) $a^{-1}$ in the multi-step oxidation mechanisms based on naphthalene. Similarly, for toluene, changes in the oxidation rate (-33 %) are still outweighed by accounting for the





high-yield $HO_2$ pathway, such that the global annual-total SOA production rate changes by 15.5 Tg (SOA) $a^{-1}$ (or 85 %), from 18.4 Tg (SOA) $a^{-1}$ in the single step oxidation mechanism based on α-pinene, to 34.0 Tg (SOA) $a^{-1}$ in the multi-step oxidation mechanisms based on toluene. However, for the case of benzene, the substantial change in oxidation rate (-66 %) is not outweighed by accounting for the difference in volatility between $RO_2$ reactions, such that the global annual-total SOA

production rate changes by -0.5 Tg (SOA) $a^{-1}$ (or -3 %), from 18.4 Tg (SOA) $a^{-1}$ in the single step oxidation mechanism based on α-pinene, to 17.9 Tg (SOA) $a^{-1}$ in the multi-step oxidation mechanisms based on benzene. Therefore, from a global perspective, the net effects of increased reaction steps and accounting for the influence of NOx on reaction yields, can either increase (85 – 150 %) of reduce (-3 %) SOA production depending on the assumed chemical reactivity of the parent VOC.

These variations in oxidation mechanisms can either improve or worsen model agreement with observations,

depending on the chemical reactivity of the parent VOC. For the single-step oxidation mechanism with a fixed reaction yield of 13 % and reactivity based on α-pinene, the model underestimated SOA across northern hemisphere mid-latitudes, with an NMB of -66 %. However, for multi-generation oxidation mechanisms with varying reaction yields, and reactivity based on either naphthalene, toluene or benzene, the NMB across northern hemisphere mid-latitudes is either -46, -56 or -71 %, respectively. These results highlight how, increases to reaction intermediates and accounting for the influence of NOx, has the

ability to both improve and worsen model agreement with observations which, crucially, depends on the assumed chemical reactivity of the parent VOC. Global annual-total emissions of benzne and toluene are 5.6 and and 6.9 Tg (C) $a^{-1}$, resptively (Henze et al., 2008), whereas emissions of naphthalene are 0.22 Tg (C) $a^{-1}$ (Pye and Seinfeld, 2010). This suggests benzene and tolune could be more relasitc surrogate compounds to represent $VOC_{ANT/BB}$ mchemistry, as opposed to naphthalene.

These results highlight that the global SOA budget is highly sensitive to hydrocarbon physicochemical processes. For

example, the global annual-total SOA production rate has varied from 47 to 75 Tg (SOA) $a^{-1}$ due to variations in VOC deposition. The global annual-total SOA production rate from anthropogenic and biomass burning emissions has varied from 17.9 to 45.1 Tg (SOA) $a^{-1}$ due to variations in VOC oxidation mechanisms. The lowest estimate of the global annual-total SOA production rate from this study would result from the combination of including precursor deposition with the multi-step oxidation pathway with reactivity of benzene. The highest estimate of the global annual-total SOA production rate from this

study would comprise of assuming no precursor deposition, but with anthropogenic and biomass burning hydrocarbons undergoing a multi-step oxidation with reactivity based on naphthalene.

Despite the limitations of this study, such as the lack of chemical complexity and geographical coverage of observations, it is apparent that SOA precursor deposition and oxidation contribute considerably towards uncertainties in both the global SOA budget and model agreement with observations. These results highlight the need for greater insight into the

physicochemical processes of gas-phase hydrocarbons related to SOA production, together with a greater density of observations.



**Code and data availability**

The model used in this study is the Global Atmosphere 4.0 (GA4.0) configuration of the HadGEM3 climate model with interactive chemistry and aerosols from UKCA, both of which are based on the UK Met Office's Unified Model (UM). Due to intellectual property right restrictions, we cannot provide either the source code or documentation papers for the UM. The Met Office Unified Model is available for use under licence. A number of research organizations and national meteorological services use the UM in collaboration with the Met Office to undertake basic atmospheric process research, produce forecasts, develop the UM code and build and evaluate Earth system models. For further information on how to apply for a licence, see http://www.metoffice.gov.uk/research/modelling-systems/unified-model. Observations from the AMS dataset can be assessed by https://sites.google.com/site/amsglobaldatabase/home (last accessed 24/05/2018).

*Acknowledgements.* This work is supported by Natural Environment Research Council (NERC; NE/L008947/1) and the Met Office through a CASE award. The development of UKCA and Fiona M. O'Connor are supported by the Joint UK BEIS/Defra Met Office Hadley Centre Climate Programme (GA01101). F.M. O'Connor also acknowledges additional funding received from the Horizon 2020 European Union's Framework Programme for Research and Innovation "Coordinated Research in Earth Systems and Climate: Experiments, Knowledge, Dissemination and Outreach (CRESCENDO)" project under grant agreement no. 641816. Computer resources provided by the Met Office, the MONSooN supercomputer facility, were used for the UKCA simulations reported here. The MONSooN system is a collaborative facility supplied under the Joint Weather and Climate Research Programme (JWCRP), which is a strategic partnership between the Met Office and NERC. ERA-Interim data used in this study were provided by the European Centre for Medium Range Weather Forecasts (ECMWF).

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

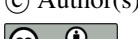



**Table 1** – Kinetic parameters used to calculate rate coefficient (Eq (1)) for both existing and new SOA precursors, taken from

5    Atkinson and Arey (2003).

| Reaction | $k_0$ / $10^{-12}$ x $cm^3$ molecule$^{-1}$ s$^{-1}$ | B / K | k (298) / $10^{-12}$ x $cm^3$ molecule$^{-1}$ s$^{-1}$ | Stoichiometric yield / % |
|---|---|---|---|---|
| | existing reaction kinetics | | | |
| monoterpene+ OH | 12.0 | -444.0 | 52.9 | 13 |
| monoterpene+ $O_3$ | 0.00101 | 732.0 | 0.0000862 | 13 |
| monoterpene+ $NO_3$ | 1.19 | -925.0 | 6.12 | 13 |
| isoprene + OH | 27.0 | -390.0 | 99.3 | 13 |
| isoprene + $O_3$ | 0.01 | 1195.0 | 0.000180 | 13 |
| isoprene + $NO_3$ | 3.15 | 450.0 | 0.692 | 13 |
| | new reaction kinetics | | | |
| naphthalene + OH | 15.7 | -117.0 | 23.2 | 100 |
| toluene + OH | 1.82 | -338.0 | 5.62 | 100 |
| benzene + OH | 2.34 | 193.0 | 1.22 | 100 |
| $RO_2$ + $HO_2$ | 1.41 | -700.0 | 14.7 | See table 3 |
| $RO_2$+ NO | 2.62 | -350.0 | 8.42 | See table 3 |





**Table 2** – Surface resistances for SOA precursors over the 9 different surface types in the model.  'Low' represents surface
resistances of ROOH, which were derived from field studies (Hall et al., 1999;Nguyen et al., 2015). 'High' represents surface
resistances of CO.

| surface type | Surface resistance ($r_c$) / sm$^{-1}$ | |
| --- | --- | --- |
| | Low | High |
| Broadleaf trees | 30 | 3700 |
| Needleleaf trees | 10 | 7300 |
| C3 grasses | 10 | 4550 |
| C4 grasses | 10 | 1960 |
| Shrubs | 10 | 4550 |
| Urban | 10 | - |
| Water | 10 | - |
| Bare soil | 10 | 4550 |
| Ice | 20000 | - |





**Table 3** – Simulations conducted in this study. Surface resistances, Low and High, are shown in Table 2. For both surface
5   resistances and H$_{eff}$, all SOA precursors are assumed to have identical parameters. The oxidation mechanism for isoprene and
monoterpene follows Eq (8) in all simulations. Emissions for all SOA precursors are identical across all simulations.

| | Surface resistance profile | H$_{eff}$ / M atm$^{-1}$ | VOC$_{ANT/BB}$ oxidation mechanism | $k_{VOC_{ANT/BB}+OH}$(298 K) / $10^{-12}$ x cm$^3$ molecule$^{-1}$ s$^{-1}$ | $\alpha_{RO_2+HO_2}$ / % | $\alpha_{RO_2+NO}$ / % | Global annual-total SOA production / Tg (SOA) a$^{-1}$ |
|---|---|---|---|---|---|---|---|
| Control | - | - | Eq (8) | 52.87 | - | - | 75 |
| Dry_High | Weak | - | Eq (8) | 52.87 | - | - | 51 |
| Dry_Low | Strong | - | Eq (8) | 52.87 | - | - | 73 |
| Wet_Low | - | $10^5$ | Eq (8) | 52.87 | - | - | 63 |
| Wet_High | - | $10^9$ | Eq (8) | 52.87 | - | - | 62 |
| DryH_WetL | Weak | $10^5$ | Eq (8) | 52.87 | - | - | 47 |
| Multi_nap | Weak | $10^5$ | Eq (9) | 23.32 | 13 | 13 | 46 |
| Multi_nap_yield | Weak | $10^5$ | Eq (9) | 23.32 | 66 | 13 | 71 |
| Multi_tol_yield | Weak | $10^5$ | Eq (9) | 5.66 | 66 | 13 | 61 |
| Multi_benz_yield | Weak | $10^5$ | Eq (9) | 1.22 | 66 | 13 | 46 |



**Figure 1** – Formation of lower volatility vapours from toluene photooxidation, as described in Ng et al. (2007).





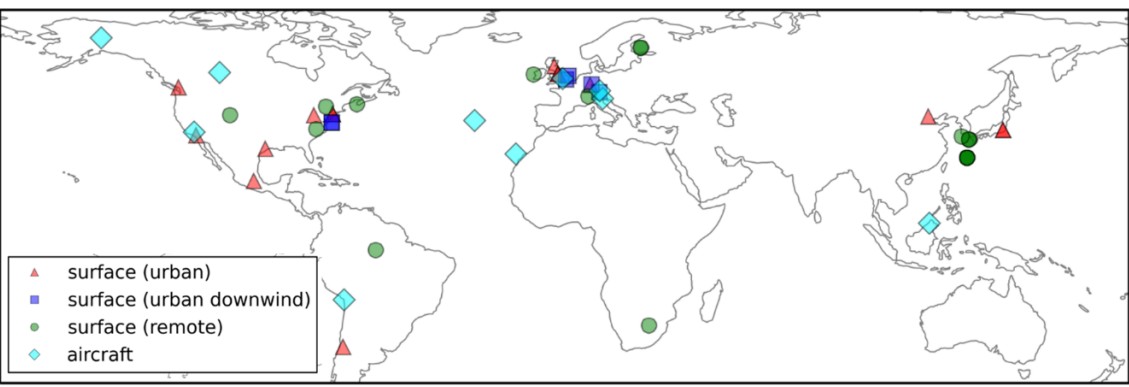

**Figure 2 –** Global map showing the 40 surface AMS observations, originally compiled by Zhang et al. (2007) and classified as urban (red triangles), urban downwind (blue squares) or remote (green circles). Of the surface observations, 37 have been classified as hydrocarbon-like OA and oxygenated-OA. Observations from 10 aircraft campaigns, originally compiled by Heald et al. (2011), are also shown (light blue diamonds). These remain as total OA.




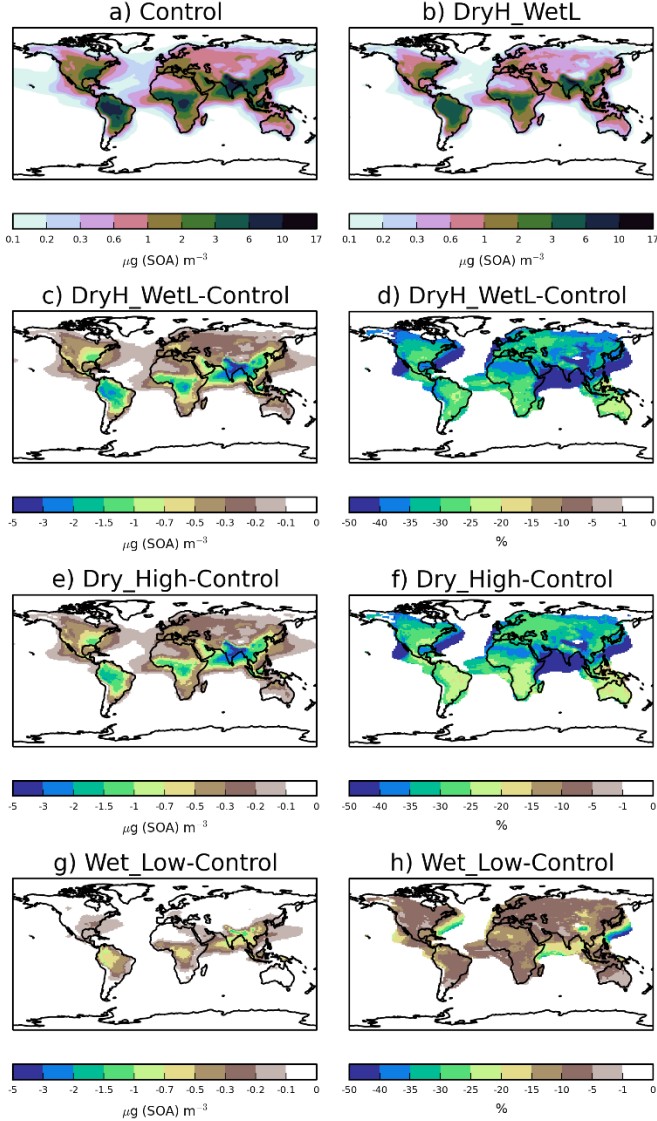

**Figure 3** – Annual-average surface SOA concentrations for a) Control, and b) DryH_WetL simulations, and absolute and percentage differences in annual-average surface SOA concentrations for (c - d) DryH_WetL, (e - f) Wet_Low, and (h – i) Dry_High simulations relative to the Control.



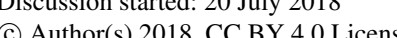

**Figure 4** – Simulated versus observed surface SOA concentrations (µg m$^{-3}$) for a) Control, b) Dry_High, c) Wet_Low and d) DryH_WetL simulations, described in Table 3. Observations, originally compiled by Zhang et al. (2007), for the time period 2000-2010, are classified by site type - urban (blue), urban downwind (green) or remote (red), and continent – Asia (squares), North America (circles) and Europe (triangles). Observed oxygenated-OA is assumed to be comparable to simulated SOA.





The 1:1 (solid), 1:2 and 2:1 (dashed), and 1:10 and 10:1 (dotted) lines are indicated. Numerical values in the bottom right of each panel indicate the normalised mean bias (%).





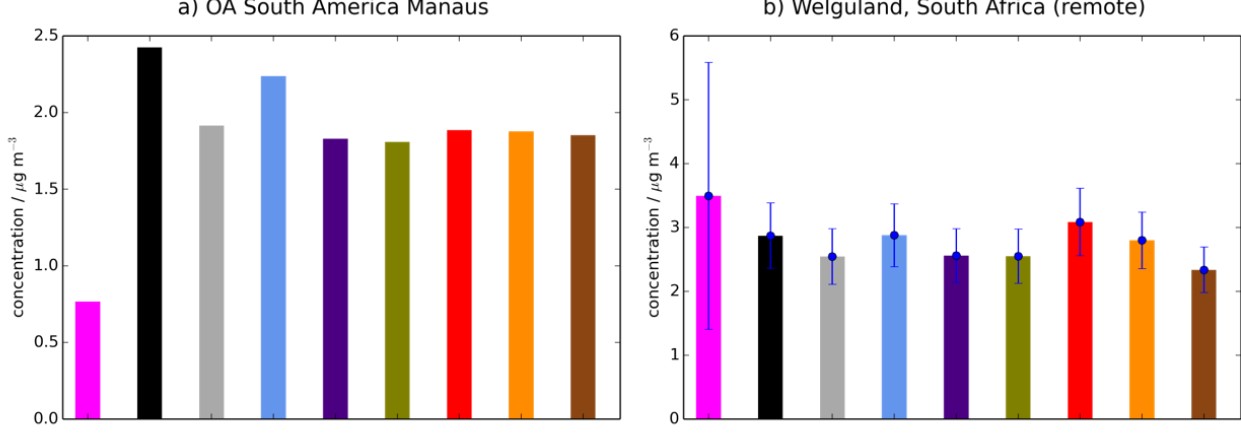

**Figure 5** – Simulated and observed OA surface concentrations (µg m$^{-3}$) over remote sites in the SH, a) Manaus (Brazil), and b) Welgegund (South Africa). Bars indicate OA concentrations from observed (pink), and simulated from the Control (black), Dry_High (grey), Wet_Low (blue), DryH_WetL (purple), Multi_nap (green), Multi_nap_yield (red), Multi_tol_yield (yellow), and Multi_benz_yield (brown) simulations, described in Table 3. For Welgegund, both observed and simulated monthly mean OA concentrations span an entire year. The standard deviations across this year, based on the monthly-mean data, are indicated in blue. For Manaus however, the measurements of OA only span one month, hence, no standard deviation is shown for this site.





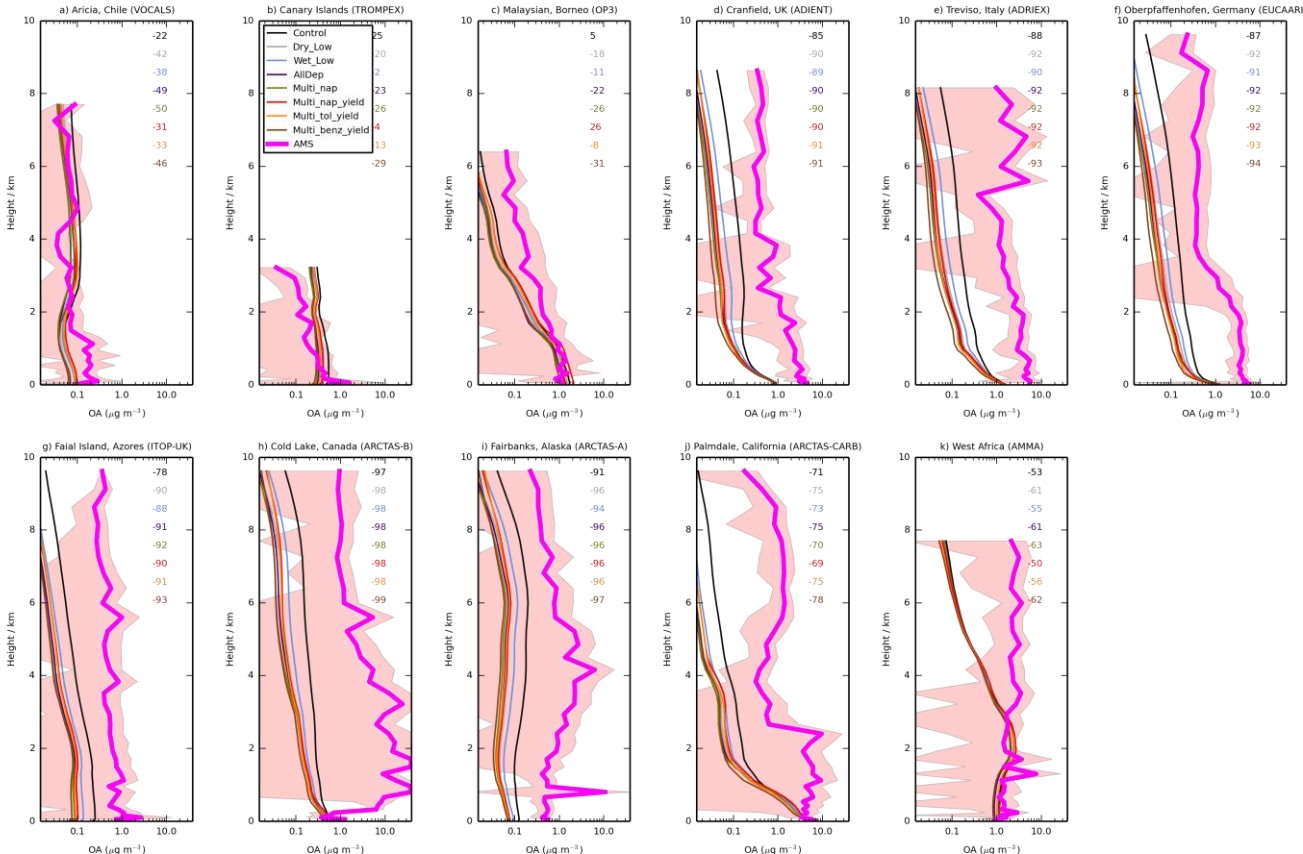

**Figure 6** – Mean vertical profile of OA (µg m⁻³) from 11 field campaigns (pink) with monthly mean modelled OA from UKCA for the simulations described in Table 3. The standard deviation of the binned observations at each model layer is shown (peach envelope). For each campaign, the normalised mean bias (%) for each simulation is also included in the top right of each panel.





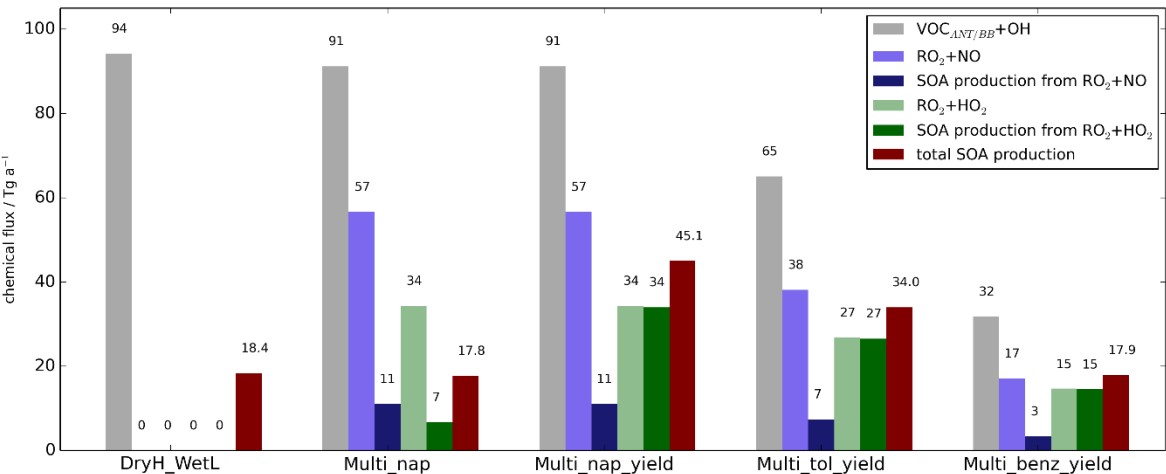

**Figure 7** – Global annual-total reaction fluxes and total SOA production rate from anthropogenic and biomass burning hydrocarbons, for the simulations described in Table 3. The global annual-total VOC$_{ANT/BB}$ emission rate, of 176 (VOC$_{ANT/BB}$) a$^{-1}$, is identical across all simulations.





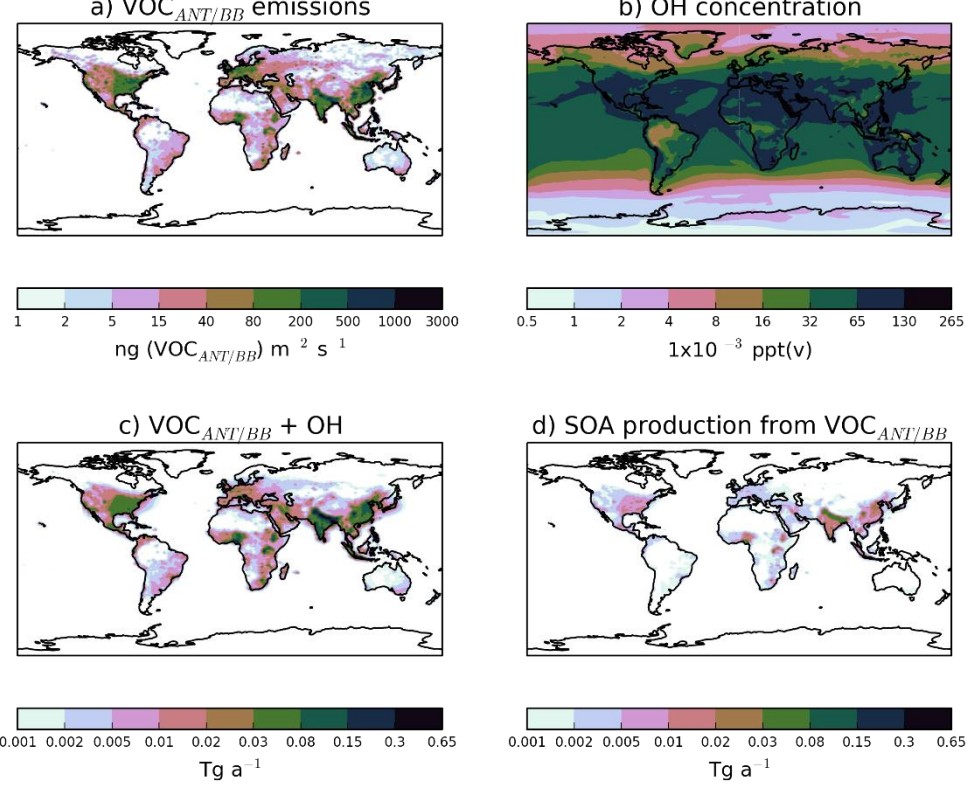

**Figure 8** – Global distributions of a) the annual-total VOC$_{ANT/BB}$ emission rate (ng (VOC$_{ANT/BB}$) m$^{-2}$ s$^{-2}$), b) the annual mean surface OH concentrations (ppq(v)), c) the annual-total vertically integrated VOC$_{ANT/BB}$ oxidation rate by OH (Tg a$^{-1}$), and d) the corresponding annual-total SOA production rate (Tg a$^{-1}$), when SOA precursor deposition and a single oxidation step with a yield of 13 % is applied (DryH_WetL; Table 3).



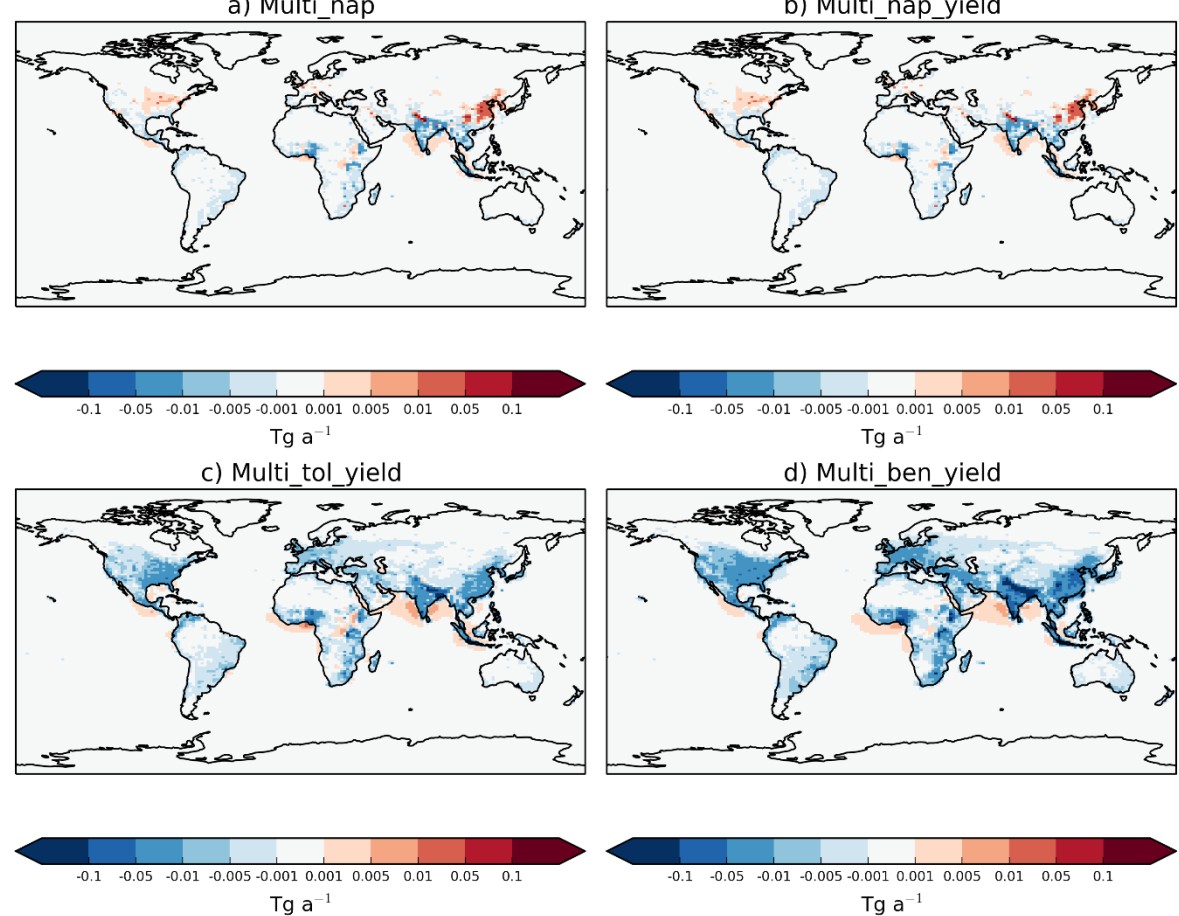

**Figure 9** – Global distribution of the absolute differences in annual-total vertically integrated VOC$_{ANT/BB}$ oxidation rates (Tg (VOC) a$^{-1}$) in a) the Multi_nap, b) the Multi_nap_yield, c) the Multi_tol_yield, and d) the Multi_ben_yield simulations relative to the DryH_WetL simulation.




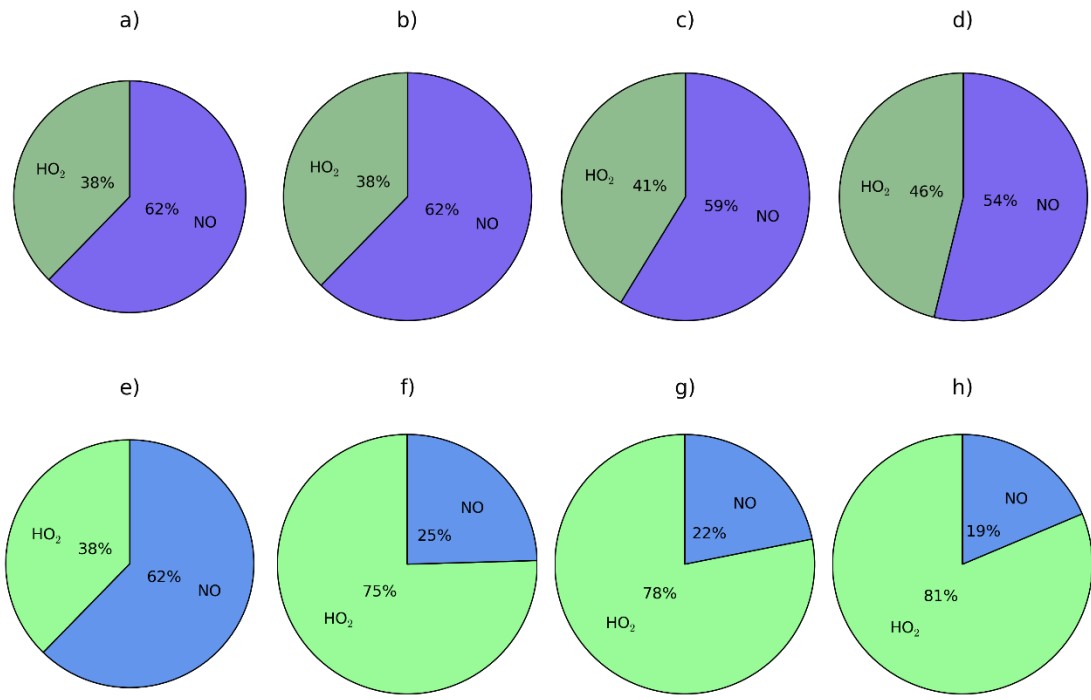

**Figure 10** – For the peroxy radical, chemical removal (top row; a – d) and SOA production (bottom row; e – h) for the Multi_nap (first column; a and e), Multi_nap_yield (second column; b and f), Multi_tol_yield (third column; c and g), and Multi_benz_yield (fourth column; d and h) simulations, which are described in Table 3.





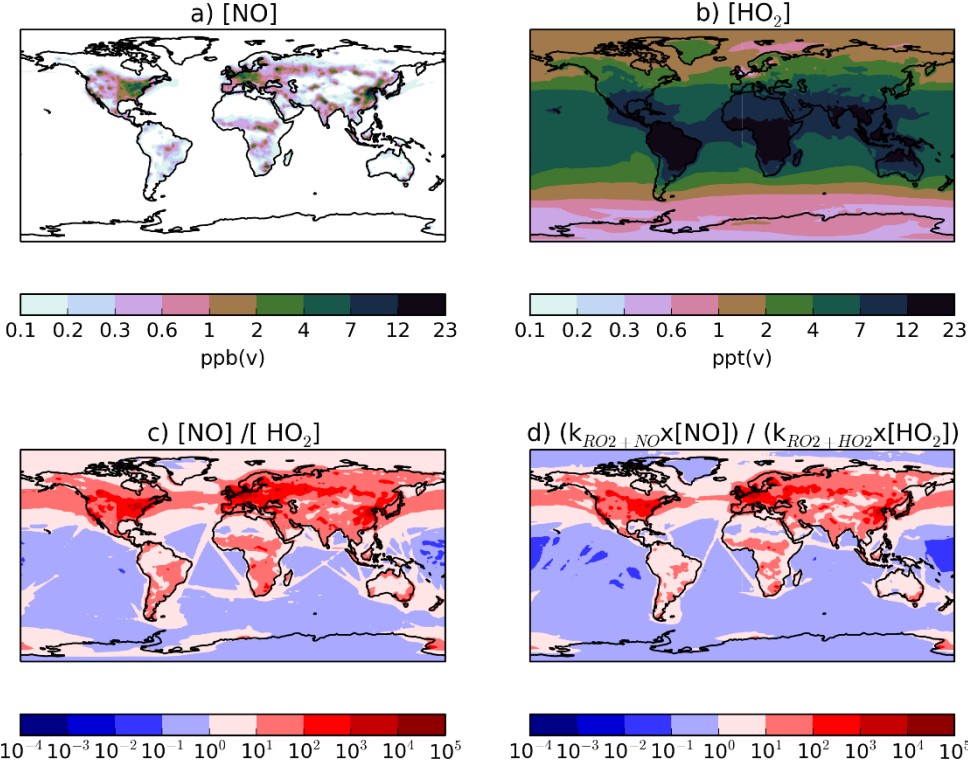

5  **Figure 11** – Global distributions of annual-average (a) surface NO concentrations (ppb(v)), (b) surface $HO_2$ concentrations (ppt(v)), c) the ratio of surface $NO/HO_2$, and d) the ratio of surface $(k_{RO2+NO}$ x NO$)/(k_{RO2+HO2}$ x $HO_2)$, where k represents the rate coefficient at 298 K. Note that the concentrations of the $HO_2$ radical are in units of ppt(v), whereas NO is in units of ppb(v).





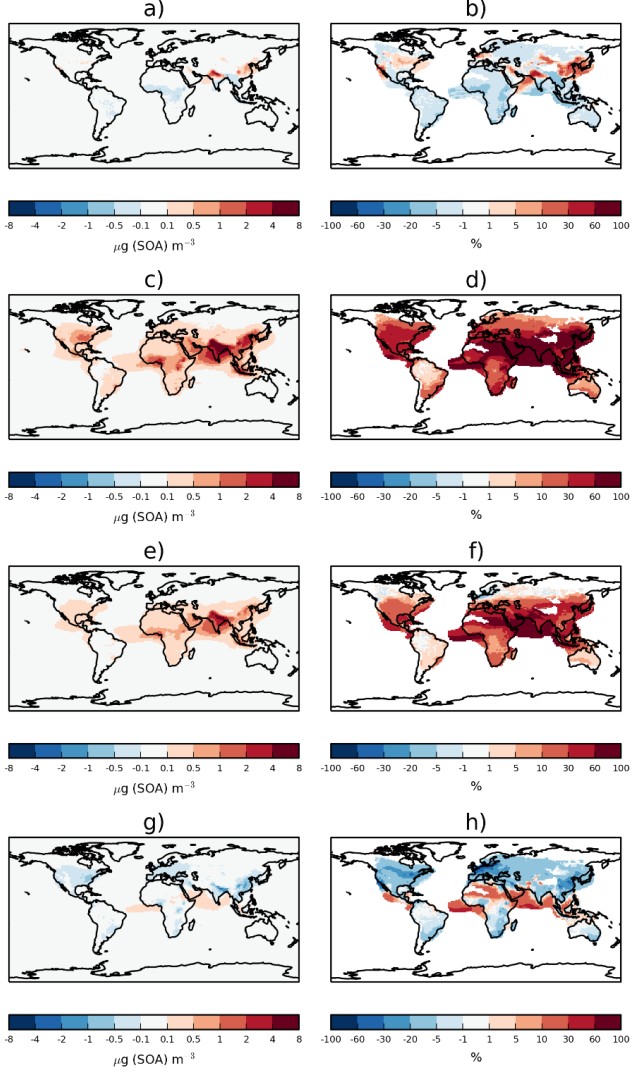

**Figure 12** – Difference in annual-average surface SOA concentrations, expressed as absolute concentrations (µg m⁻³) (left column) and as a percentage (right column) between Multi_nap (top row; a - b), Multi_nap_yield (second row; c - d), Multi_tol_yield (third row; e - f), and Multi_benz_yield (fourth row; g - h) and the DryH_WetL simulation, which are all described in Table 3.





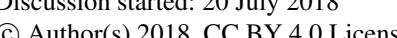

Figure 13 – Simulated versus observed SOA concentrations (µg m$^{-3}$) for a) Multi_nap, b) Multi_nap_yield c) Multi_tol_yield and d) Multi_benz_yield simulations, described in Table 3. Observations for the time period 2000-, are classified by site type - urban (blue), urban downwind (green) or remote (red), and continent – Asia (squares), North America (circles) and Europe (triangles). Observed oxygenated-OA is assumed to be comparable to simulated SOA. The 1:1 (solid), 1:2 and 2:1 (dashed),



and 1:10 and 10:1 (dotted) lines are indicated. Numerical values in the bottom right of each panel indicate the normalised mean bias (%).

