# Peer review of "The roles of volatile organic compound deposition and oxidation mechanisms in determining secondary organic aerosol production: A global perspective using the UKCA chemistry-climate model (vn8.4)"

_Geoscientific Model Development, 2018_

## Referee Comment (RC1) · Anonymous Referee #1 · 22 Aug 2018

General Comments

This work expands the SOA description in the United Kingdom Chemistry and Aerosol (UKCA) chemistry-climate model, and adequately explains why there is need for a more complex description of SOA formation in UKCA. The work also compares UKCA against global observations reasonably well. However, major revisions in the design of the model set-up and interpretation of the results are needed. These changes are

more explicitly stated in the specific comments, but generally described here. With these changes, the work has a potential to make a nice contribution to the field.

-There is no mention of the higher SOA yields for toluene measured in Zhang et al., 2014 paper. This paper determines that when chamber vapor wall loss effects are accounted for, the toluene SOA yields increase significantly for both the RO2 + NO and RO2 + HO2 channels compared to previous studies. The work mostly cites a paper (Ng 2007) from the same group, but 7 years prior. The results in Zhang et al., 2014 paper should be considered in this work and used to form a basis for the sensitivity tests that are performed.

Xuan Zhang, Christopher D. Cappa, Shantanu H. Jathar, Renee C. McVay, Joseph J. Ensberg, Michael J. Kleeman, and John H. Seinfeld: Influence of vapor wall loss in laboratory chambers on yields of secondary organic aerosol, 111 (16), 5802-5807, doi: https://doi.org/10.1073/pnas.1404727111, 2015.

-There is a lot of discussion about how in the aromatic system the RO2 + NO pathway forms semi-volatile compounds and the RO2 + HO2 pathway forms non-volatile compounds. This work does not benefit from such a discussion. The SOA gas surrogate species irreversibly forms SOA in the model used in this work in all simulations. The work is misleading to suggest that the difference in volatility is accounted for by increasing the SOA molar yield. To account for volatility a surrogate species that will reversibly partition to the particle-phase based on its volatility is required. Accurately representing this process has different consequences than increasing the SOA yield. More commonly and possibly more applicable to this study, the RO2 + HO2 products are seen as more functionalized with a higher SOA yield and RO2 + NO products are seen as more fragmented with a lower SOA yield.

-Throughout the work, the advances to the SOA scheme are labeled as multigenerational. This is very misleading. Multigenerational typically does not include the peroxy radical as another generation. For example, A + OH -> RO2; RO2 + NO -> Organic

nitrate. This organic nitrate as the first non-radical stable product, is a first-generation product. If Organic nitrate + OH-> products is added, this is a multigenerational set-up. For consistency, with past work and the general use of multigenerational in the field, I would suggest changing this to RO2 fate throughout this work. Also the work spends a lot of time discussing how adding the RO2 radical step may delay SOA formation and states that the RO2 radical has a lifetime with respect to oxidation of $\sim$ 1 day. This should be verified. For example, a good recommended source for describing RO2 oxidation in the atmosphere, Orlando et al. 2012, suggests at most this RO2 lifetime is many minutes.

John Orlando and Geoffrey Tyndall: Laboratory studies of organic peroxy radical chemistry: an overview with emphasis on recent issues of atmospheric significance, Chemical Society Reviews, 41, 6294-6317, doi: 10.1039/C2CS35166H, 2012.

-Overall and as explained in the specific comments below there is not sufficient justification for why the test cases were chosen.

Specific Comments

-Page 4 line 21: "As aromatic oxidation is initiated by the hydroxy radical, the influence of NOx on SOA production is probably due to reaction of NO with second or later generation oxidation products" -> Please rephrase this, see general comment. The peroxy radical is not a second generation product.

-Page 4 line 23 and Figure 1: "Oxidation of the parent aromatic hydrocarbon . . .forming a bicyclic peroxy radical, RO2" The bicyclic peroxy radical is only the dominant mechanism for OH oxidation of an aromatic compound, there are other pathways too. The Johnson et al. 2004 paper that is cited does not explain the formation of this bicyclic peroxy radical. Johnson et al. 2004 discusses alkane alkoxy radicals. There are many sources to cite here (e.g., Birdsall 2010), who first measured the bicyclic peroxy radical. Birdsall also discusses the full chemistry that occurs for aromatics.

Adam W. Birdsall, John F. Andreoni, and Matthew J. Elrod: Investigation of the Role of Bicyclic Peroxy Radicals in the Oxidation Mechanism of Toluene, J. Phys. Chem. A., 114, 10655-10663, doi: 10.1021/jp105467e, 2010.

-page 8 line 11: How does condensation aging relate to the previous sentence? Are there additional aging processes in the model? Is there any aging of the SOA?

-page 8 line 27: "into the insoluble mode and transferred into the insoluble" Please clarify/rephrase?

-page 9 line 22: Please clarify this first sentence. What else would produce SOA other than VOCs in the model?

-page 9 section 2.5: Please clarify. Does VOCant/bb only undergo OH oxidation in the default and the updated mechanism? Perhaps, adding this to Table 1 would be useful. Are the VOCant/bb assumptions for reactivity determined in this work or another work? Further description on why these assumptions were made would be useful.

-page 10 line 5: Are the parent hydrocarbons because they are SOA precursors also wet and dry deposited using the high henry's law constants (> 10ˆ5)? Parent hydrocarbons like isoprene are well established to have lower henry's law constants. Is there only one tracer for SOG?

-page 10 line 9: Why use a henry's law constant range for wet deposition and not also test the same range for dry deposition? This seems like a more consistent approach and more fairly capturing the actual uncertainties. Using the experimentally observed surface resistances does seem reasonable for the ROOH as SOA precursors are likely to have hydroperoxy groups. However, why bound the uncertainty with CO? Are SOA precursors expected to act similarly to CO? This likely adds extra unnecessary uncertainty to the model results.

Page 11 line 13: See general comment above. It is misleading to suggest that increasing the SOA molar yield will account for the difference in volatility between different

products. To account for volatility differences in a model you must have SOA precursors that are able to reversibly partition to the particle phase.

Page 12 line 18: "RO2 and SOG have differing relative molecular masses. Consequently, a stochiometric yield of 66% corresponds to a mass yield of 100%. Therefore, 66% is the highest stoichiometric yield that ensures conservation of mass without the addition of other atoms, such as oxygen" Please clarify. The logic here seems incorrect. First, why choose the highest SOA yield possible? Why not use an SOA yield measured/constrained from experimental studies (e.g., the SOA yields measured by Zhang et al., 2014)? Second, why is mass conservation necessary. Although this reaction is written as one step it is really a parameterization of many reactions and so does not need to follow the laws of mass conservation. Although very unlikely, the highest SOA yield possible is unity molar yield from the parent VOC molecule. The same example I used above. A + OH -> R; R + O2 -> RO2; RO2 + NO -> Organic nitrate. This organic nitrate has a lot more mass than the parent molecule A, because it is more functionalized and has gained oxygen and nitrogen atoms by reacting with OH, O2, and NO.

-Page 13 line 22: Were the model and observations compared separately in 2000, so that there would be comparisons over the same year the model was run? How do these 2000 results compare to the 2000-2010 more general results? From 2000-2010 there are substantial changes in anthropogenic and fire emissions, which would make it difficult to interpret these results.

Figure 4 and Figure 13: What is the averaging for the model and observations used to get these points? Was any seasonal analysis conducted? Are these points a mix of different seasons?

-Page 14: Please add some explanation in the paragraphs below or elsewhere in the paper about how this work might differ between past work. Not necessarily in overall magnitudes, but in approach. For example, this work uses SOA precursors that

irreversibly form SOA, while past work has used SOA precursors that reversibly form SOA (e.g., volatility basis set). Explain how this might affect the results in this work especially the impact of wet/dry deposition?

-Section 5: This section would be much more effective if it were written more concisely.

-Page 19 line 12: See general comment above. The use of multigenerational here and throughout the work is misleading. I would suggest phrasing this instead as RO2 fate.

-Page 20 line 18: The lifetime of the RO2 radical being ∼1day is quite unexpected. Please confirm this and considering the actual RO2 radical lifetime, rephrase this section.

-Page 21 line 6: Is NO actually high in the Amazon in your model? It looks low in Figure 11.

-Page 23 line 14: Increasing the molar SOA yield is not equivalent to changes in volatility. Please rephrase this and all paragraphs in this discussion. Unless you change the volatility of the SOA precursors and have SOA form reversibly you are not actually accounting for the changes in volatility.

-Page 23 line 16: See above, please reconsider/clarify why 0.66 was chosen as the SOA molar yield?

-Page 29 line 27: Please expand on this paragraph.

-Table 2: Please explain how the field studies were used to derive these surface resistance values. What would these surface resistances be, if the henry's law constants used for wet deposition were used here instead?

Technical Corrections

-NOx sometimes has x subscripted and sometimes not. -You have 2 section 2.5 - There are a number of spelling errors throughout as noted below: -page 2, line 4 (improvements), line 6 (observations) -page 6, line 13 (precursors) -page 15, line 33

(respectively) -page 18 line 12 (translated) -page 26 line 26 (respectively) -page 26 line 28 (chemistry) -page 27 line 3 (Africa)

---

## Referee Comment (RC2) · Anonymous Referee #2 · 19 Nov 2018

This work by Kelly et al. investigate the impacts of VOC deposition and oxidation mechanisms on SOA formation within the United Kingdom Chemistry and Aerosol (UKCA) model. This work evaluated simulated OA/SOA with surface and aircraft observed data in different areas around the globe. This work highlights the uncertainties in the global SOA budget associated with the changes in SOA schemes. I will suggest to accept this manuscript after minor revisions. My specific comments are listed below.

[Figure]
(1) Emissions. How may VOC from biomass burning and anthropogenic sources respectively? Are both VOCBB and VOCANT are assumed to emit from the surface? The biomass burning source could be elevate emissions and might be impact on some of the results in this study. For example, in page 21, " At higher levels, NO/HO2 reduces, suggesting an increasing importance of the HO2 pathway at higher altitudes. However, due to the fast chemical reactivity, the majority of SOA production occurs at the surface. For the majority of the atmosphere, the difference in the magnitudes of the oxidant concentrations favours the RO2 +NO pathway over the RO2 +HO2 pathway." Therefore, if the SOA production occurs in higher altitude because of elevate emissions, more SOA will produce through HO2.

(2) Default Treatment of SOA. The SOG condenses irreversibly to form SOA in UKCA. Will it lead to a different result if the model assumes the SOG condenses reversibly to form SOA?

(3) Could author discuss about the potential impacts of the precursors deposition on SOA production associated with different source types?

(4) Page 8 Line 27: "All carbonaceous primary emissions are emitted into the insoluble mode and transferred into the insoluble" I cannot understand this sentence.

(5) Page 12 Line 15-20. "Consequently, a stoichiometric yield of 66 % corresponds to a mass yield of 100%. Therefore, 66% is the highest stoichiometric yield that ensures conservation of mass without the addition of other atoms, such as oxygen." Why the mass conservation used here?

(6) Page 14. Line 20. "all VOC source ranging from 47 to 74 Tg (SOA) a-1" change to "47 to 75"?

(7) Page 15. Line 17-20. What is the lifetime of SOA in this study?

(8) Page 15. Line 30. How model predicted SOA compare to the observations? Do they use monthly averaged or median values? Since "1.875° longitude by 1.25° latitude" in

this study is a really coarse resolution, will the comparison with remote sites seems better?

(9) Page 19. Line 25-30. " these changes in annual-total VOCANT/BB oxidation rates within emissions source regions correspond to reductions between 10 and 30 % (not shown). By contrast, downwind of many emissions source regions, the lower reactivity acts to enhance VOCANT/BB oxidation rates." It is really hard for me to find the down-wind emission source regions because the largest increase occurs in source regions such as China and East US. Could the author give a map plot to point out where these downwind regions are?

(10) Page 26. Line 5-10. " Although the global annual-total SOA production rates are identical, the global annual-average SOA burden is 10 % greater when using benzene as the parent VOC undergoing multi-generational oxidation, highlighting the strong spatial gradients in SOA lifetime." Could author explain how the SOA lifetime changes?

---

## Referee Comment (RC3) · Anonymous Referee #3 · 29 Nov 2018

Secondary organic aerosol (SOA) is an important but the least understood component of atmospheric aerosols. SOA life-cycle involves many chemical and physical processes, including emission, gas-phase chemistry, aqueous/solid phase chemistry, condensation, deposition and etc. This makes the global SOA modeling really challenging. This manuscript investigated the sensitivities of SOA formation to the different volatile organic compound (VOC) deposition and oxidation mechanism use a global chemistry-climate model (UKCA). It also compared these sensitivity simulations

against the observations to see how these difference mechanisms affect the model-observation agreements. Overall, this manuscript is organized well and provide readers deep insights on how VOC deposition and oxidation reactions affect the SOA production. I recommend publishing it after the authors address my comments below.

General comments

It is not clear to me why the authors only use aromatics as the biomass burning SOA precursor. How representative are the aromatics for the biomass burning SOA precursor?

All model simulations underestimate the observed OA concentrations. The authors should at least discuss the reasons for this underestimation and its potential impact on this paper's conclusions.

Specific comments

P3, line 1. Kelly et al., 2018 is not listed in the reference list.

P3, line 7. Suggesting changing "an aspects of SOA" to "another aspect of SOA", because the previous sentence already described one aspect of SOA difference between different models.

P4, line 8-9. Can you list some references to support this statement?

P8, line 20. Section 2.4 should be section 2.3. And also please change the section 2.5 number.

P8, line 27. Please change VOCBB and VOCANT to "VOCBB" and "VOCANT" to be consistent with the rest of paper.

P8, line 27. The second "insoluble" should be "soluble".

P9, line 24-26. So the model includes both the isoprene oxidation that leads to SOA formation and the isoprene oxidation in the Mainz Isoprene Mechanism? Isn't this

double-counting the isoprene oxidation?

P11, line 20. Can the authors briefly describe the kinetics for aromatic oxidations here? So the readers don't have to read the table when reading the text.

P12, line 18. "Different molecular masses". What molecular weights are used for RO2 and SOG in the model. SOG is a lumped species, right? So how do the authors know the molecular weight of SOG?

P13, line 20-25. Did the authors account for the seasonal variation of biomass burning VOC emissions in the model (i.e. monthly change emissions)?

P18, line 15. OH can be indirectly constrained by the CH4 lifetime.

P24, line 24. "Favors the likelihood of RO2 radicals entering the high-yield HO2 pathway". Why? I don't understand the reason for that.

P26, line 30. "Figure 12", is it meant to be Figure 13?

P27, line 11-17. This argument is confusing to me. Can the authors elaborate that?

Reference list. There are some references with titles being all capital letters. Please change them.

Tabe 1. Please add the VOCANT and VOCBB oxidation kinetics in the "existing reaction kinetics" subsection.

---

## Author Comment (AC1) · 28 Feb 2019

**Responses to referee comments on "The role of volatile organic compound deposition and oxidation mechanisms in determining secondary organic aerosol production: A global perspective using the UKCA chemistry-climate model (vn8.4)" by Jamie M. Kelly et al.**

**We thank all referees for their insightful feedback that has considerably improved the manuscript. For each of the referees' comments (RC) (indicated by quotation marks), we have provided our author response (AR) and the modified text within the updated manuscript (indicated by italics). In our revised manuscript, modified text is highlighted using tracked changes.**

**Referee #1 (received on 22$^{nd}$ August 2018)**

RC1: 'This work expands the SOA description in the United Kingdom Chemistry and Aerosol (UKCA) chemistry-climate model, and adequately explains why there is need for a more complex description of SOA formation in UKCA. The work also compares UKCA against global observations reasonably well. However, major revisions in the design of the model set-up and interpretation of the results are needed. These changes are more explicitly stated in the specific comments, but generally described here. With these changes, the work has a potential to make a nice contribution to the field.'

> AC1: We thank the referee for their positive feedback on the manuscript.

RC2: 'There is no mention of the higher SOA yields for toluene measured in Zhang et al., 2014 paper. This paper determines that when chamber vapor wall loss effects are accounted for, the toluene SOA yields increase significantly for both the RO2 + NO and RO2 + HO2 channels compared to previous studies. The work mostly cites a paper (Ng 2007) from the same group, but 7 years prior. The results in Zhang et al., 2014 paper should be considered in this work and used to form a basis for the sensitivity tests that are performed.

Xuan Zhang, Christopher D. Cappa, Shantanu H. Jathar, Renee C. McVay, Joseph J. Ensberg, Michael J. Kleeman, and John H. Seinfeld: Influence of vapor wall loss in laboratory chambers on yields of secondary organic aerosol, 111 (16), 5802-5807, doi: https://doi.org/10.1073/pnas.1404727111, 2015.'

> AC2: We thank the referee for pointing this out. In our study, we are not selecting laboratory-measured SOA yields, we are only attempting to mirror the high and low SOA yield pathways for RO2 with HO2 and NO, respectively. One of the first major publications to identify and explain this behaviour was Ng et al. (2007), and that is the reason for referring to this publication throughout the manuscript. The more recent Zhang et al (2014) paper further corroborates this behaviour, whilst also highlighting how SOA yields are generally underestimated due to wall losses. We have edited the manuscript to including this missing citation.

> Page 4 lines 14-16: *Zhang et al. (2014) further corroborates this negative sensitivity of SOA yields from aromatic compounds to $NO_X$ concentrations, and also highlights how chamber studies frequently underestimate SOA yields due to wall losses.*

RC3: 'There is a lot of discussion about how in the aromatic system the RO2 + NO pathway forms semi-volatile compounds and the RO2 + HO2 pathway forms non-volatile compounds. This work does not benefit from such a discussion. The SOA gas surrogate species irreversibly forms SOA in the model used in this work in all simulations. The work is misleading to suggest that the difference in volatility is accounted for by increasing the SOA molar yield. To account for volatility a surrogate species that will reversibly partition to the particle-phase based on its volatility is required. Accurately representing this process has different consequences than increasing the SOA yield. More commonly and possibly more applicable to this study, the RO2 + HO2 products are seen as more functionalized with a higher SOA yield and RO2 + NO products are seen as more fragmented with a lower SOA yield.'

> AC3: We completely agree with the referee on this point. In the updated manuscript we have edited our description.
>
> Page 4 lines 22-24: *Under high-$NO_X$ conditions, the peroxy radical reacts with the nitric oxide radical (NO) to form fragmented products, whereas, under low-$NO_X$ conditions, the peroxy radical reacts with the hydroperoxyl radical ($HO_2$) to form functionalised products.*

RC4: 'Throughout the work, the advances to the SOA scheme are labeled as multigenerational. This is very misleading. Multigenerational typically does not include the peroxy radical as another generation. For example, A + OH -> RO2; RO2 + NO -> Organic nitrate. This organic nitrate as the first non-radical stable product, is a first-generation product. If Organic nitrate + OH-> products is added, this is a multigenerational set-up. For consistency, with past work and the general use of multigenerational in the field, I would suggest changing this to RO2 fate throughout this work. Also the work spends a lot of time discussing how adding the RO2 radical step may delay SOA formation and states that the RO2 radical has a lifetime with respect to oxidation of ~ 1 day. This should be verified. For example, a good recommended source for describing RO2 oxidation in the atmosphere, Orlando et al. 2012, suggests at most this RO2 lifetime is many minutes.

John Orlando and Geoffrey Tyndall: Laboratory studies of organic peroxy radical chemistry: an overview with emphasis on recent issues of atmospheric significance, Chemical Society Reviews, 41, 6294-6317, doi: 10.1039/C2CS35166H, 2012.'

> AC4: We agree that referring to the oxidation mechanisms which include a reaction intermediate as 'multigenerational' is misleading. In the updated manuscript, we have replaced 'single-step oxidation mechanisms' with 'oxidation mechanisms with no reaction intermediate', and we have replaced 'multigenerational oxidation mechanisms' with 'oxidation mechanisms with the reaction intermediate'. With respect to $RO_2$, we do not assign a lifetime to $RO_2$. Instead, the lifetime is a result of the rate coefficient (which is from a published study) and oxidant availability. So differences in $RO_2$ lifetime could be due to differences in rate coefficients of oxidants. We have added the Orlando et al. (2012) citation to our discussion in the updated manuscript.
>
> Page 21 lines 3-3: Note, a review of laboratory studies suggests the lifetime of $RO_2$ could be of the order of minutes (Orlando et al., 2012).

RC5: 'Overall and as explained in the specific comments below there is not sufficient justification for why the test cases were chosen.'

AC5: We thank the referee for this comment. We have decided to respond to the specific comments below.

RC6: 'Page 4 line 21: "As aromatic oxidation is initiated by the hydroxy radical, the influence of NOx on SOA production is probably due to reaction of NO with second or later generation oxidation products" -> Please rephrase this, see general comment. The peroxy radical is not a second generation product.'

AC: Thank you for pointing this out. This has now been corrected.

Page 4 line 20: *subsequent reaction intermediates or products*

RC7: 'Page 4 line 23 and Figure 1: "Oxidation of the parent aromatic hydrocarbon . . .forming a bicyclic peroxy radical, RO2" The bicyclic peroxy radical is only the dominant mechanism for OH oxidation of an aromatic compound, there are other pathways too. The Johnson et al. 2004 paper that is cited does not explain the formation of this bicyclic peroxy radical. Johnson et al. 2004 discusses alkane alkoxy radicals. There are many sources to cite here (e.g., Birdsall 2010), who first measured the bicyclic peroxy radical. Birdsall also discusses the full chemistry that occurs for aromatics.

Adam W. Birdsall, John F. Andreoni, and Matthew J. Elrod: Investigation of the Role of Bicyclic Peroxy Radicals in the Oxidation Mechanism of Toluene, J. Phys. Chem. A., 114, 10655-10663, doi: 10.1021/jp105467e, 2010.'

AC7: We thank the referee for informing us on this mistake. We have replaced the Johnson et al (2004) citation with Birdsall et al. (2010).

Page 4 line 22; *(Koch et al., 2007; Birdsall et al., 2010).*

Page 32 lines 14-16; *Adam W. Birdsall, John F. Andreoni, and Matthew J. Elrod: Investigation of the Role of Bicyclic Peroxy Radicals in the Oxidation Mechanism of Toluene, J. Phys. Chem. A., 114, 10655-10663, doi: 10.1021/jp105467e, 2010.*

RC8: 'page 8 line 11: How does condensation aging relate to the previous sentence? Are there additional aging processes in the model? Is there any aging of the SOA?'

AC8: We thank the referee for pointing this out. We have revised this text to make it clearer.

Page 7 lines 27-32; *Aerosol microphysical processes included are nucleation, coagulation, condensation, condensation ageing, hygroscopic growth and cloud processing. Species such as POA and BC are assumed to be emitted in insoluble forms. Condensation ageing refers to soluble vapours condensing on these insoluble POA and BC particles, and thus, rendering them soluble. No aging processes are applied to SOA.*

RC9: page 8 line 27: "into the insoluble mode and transferred into the insoluble" Please clarify/rephrase?'

AC9: We have removed this text (see response to comment above).

RC10: 'page 9 line 22: Please clarify this first sentence. What else would produce SOA other than VOCs in the model?'

AC10: In response to this comment, we have clarified the text in the updated manuscript.

Page 9 lines 3-4: *In this study, SOA production is considered from gas-to-particle partitioning of VOC oxidation products. S/IVOCs emissions are not considered and aqueous phase SOA production is not included.*

RC11: 'page 9 section 2.5: Please clarify. Does VOCant/bb only undergo OH oxidation in the default and the updated mechanism? Perhaps, adding this to Table 1 would be useful. Are the VOCant/bb assumptions for reactivity determined in this work or another work? Further description on why these assumptions were made would be useful.'

AC11: Indeed, $VOC_{ANT/BB}$ is only oxidised by OH due to the assumption that it is a reduced compound. This is stated within the main body of text on page X lines X-X, 'Initially, the assumption is made that $VOC_{ANT}$ and $VOC_{BB}$ are reduced compounds, with only single carbon bonding and react predominantly with OH. $VOC_{ANT}$ and $VOC_{BB}$ are also assumed to have a similar reactivity to monoterpene towards OH oxidation, but do not react with $O_3$ or $NO_3$.' We agree with the referee that it is a good idea to reiterate these assumptions in the caption for Table 1.

Page 44 lines 5-6; *Note, $VOC_{ANT/BB}$ reacts with OH, with reaction kinetics based off either monoterpene, naphthalane, toluene or benzene.*

RC12: 'page 10 line 5: Are the parent hydrocarbons because they are SOA precursors also wet and dry deposited using the high henry's law constants (> 10^5)? Parent hydrocarbons like isoprene are well established to have lower henry's law constants. Is there only one tracer for SOG?'

AC12: I'm not entirely sure of the first question, but I will try to explain the set up. In this study, SOA precursors include the parent hydrocarbons (monoterpene, isoprene, $VOC_{ANT}$ and $VOC_{BB}$) and the condensable oxidation product (SOG). When the SOA precursors are assumed to be susceptible to either wet or dry deposition, a single effective Henry's coefficient (for wet deposition) or surface resistance coefficient (for dry deposition) is assigned to all species. We agree with the referee that parameters such as the Henry's coefficient (and surface resistances) are likely species-specific. Whilst we would be able to assign a species-specific Henry's coefficient for isoprene, we would not be able to do so for monoterpenes, $VOC_{ANT}$, $VOC_{BB}$, or SOG as they are all surrogate compounds representing a mixture of species. Therefore, we believe that, as a first attempt, it is safe to assign identical

deposition parameters across all the SOA precursors. Furthermore, our sensitivity simulations indicate how the strength of SOA production is rather insensitive to changes in the Henry's of several orders of magnitude. This implies that the inclusion of species-specific Henry's coefficient would not have significant effects on simulated SOA.

RC13: 'page 10 line 9: Why use a henry's law constant range for wet deposition and not also test the same range for dry deposition? This seems like a more consistent approach and more fairly capturing the actual uncertainties. Using the experimentally observed surface resistances does seem reasonable for the ROOH as SOA precursors are likely to have hydroperoxy groups. However, why bound the uncertainty with CO? Are SOA precursors expected to act similarly to CO? This likely adds extra unnecessary uncertainty to the model results.'

> AC13: This is a really interesting point. The Henry's coefficients are included in wet deposition calculations but not dry deposition calculations. Therefore, we cannot use the same range of Henry's coefficients to bound both these processes. ROOH was chosen as it likely has a similar structure and reactivity to our SOA precursors. CO was chosen as, despite it having a dissimilar structure and reactivity to our SOA precursors, the deposition of these species has been studied extensively, and we therefore have a high certainty in the accuracy of CO deposition parameters.

RC14: 'Page 11 line 13: See general comment above. It is misleading to suggest that increasing the SOA molar yield will account for the difference in volatility between different products. To account for volatility differences in a model you must have SOA precursors that are able to reversibly partition to the particle phase.'

> AC14: We agree with the referee that our description is inaccurate. We have revised the text in line with this discussion.

> Page 11 lines 12-15: *Both $RO_2$ reactions form the same non-volatile species, SOG, but the yields associated with the formation rates of this product are variable ($\alpha_{RO_2+HO_2}$ and $\alpha_{RO_2+NO}$). Hence, this mechanism allows the sensitivity of SOA production to $HO_2$/NO to be accounted for. However, note that the differences in volatility between $RO_2$ oxidation products are not explicitly accounted for.*

RC15: 'Page 12 line 18: "RO2 and SOG have differing relative molecular masses. Consequently, a stochiometric yield of 66% corresponds to a mass yield of 100 %. Therefore, 66% is the highest stoichiometric yield that ensures conservation of mass without the addition of other atoms, such as oxygen" Please clarify. The logic here seems incorrect. First, why choose the highest SOA yield possible? Why not use an SOA yield measured/constrained from experimental studies (e.g., the SOA yields measured by Zhang et al., 2014)? Second, why is mass conservation necessary. Although this re- action is written as one step it is really a parameterization of many reactions and so does not need to follow the laws of mass conservation. Although very unlikely, the highest SOA yield possible is unity molar yield from the parent VOC molecule. The same example I used above. A+OH->R;R+O2->RO2;RO2+NO->Organic nitrate. This organic nitrate has a lot more mass than the parent molecule A,

because it is more functionalized and has gained oxygen and nitrogen atoms by reacting with OH, O2, and NO.'

> AC15: This is an extremely interesting point the referee raises, highlighting that our objective is not entirely clear in the original manuscript. As the referee has noted, we could make use of the published laboratory-determined SOA yields for these simulations. However, these yields vary considerably from one study to another. Also, the peroxy radical in this study is a surrogate compound, representing the oxidation products a complex mixture of anthropogenic and biomass burning VOCs. Hence, it is difficult to select species-specific SOA yields from chamber studies. The sensitivity of SOA production to $NO_X$ has been identified in numerous chamber studies. The objective of this study is to test the effects of accounting for this in a global model. Hence, it is the effect of the difference in stoichiometric yields on simulated SOA which we are exploring in this study. We are not evaluating isaolted SOA yields and their corresponding SOA concentrations in absolute terms. Instead, we are quantifying a range – the Multi_nap simulations corresponds to no differences in yields, whereas the Multi_nap_yield simulation corresponds to when the yield of the RO2+HO2 pathway is 5 times higher than the RO2+NO pathway.

RC16: 'Page 13 line 22: Were the model and observations compared separately in 2000, so that there would be comparisons over the same year the model was run? How do these 2000 results compare to the 2000-2010 more general results? From 2000-2010 there are substantial changes in anthropogenic and fire emissions, which would make it difficult to interpret these results.'

> AC16: The model to measurement comparison was not conducted for observations only falling in the year 2000. We agree that this could be a good test to see how interannual variability in SOA concentrations affects our model to measurement comparison. We have noted the potential importance of this in the original manuscript, 'This mismatch in time may be particularly important for regions influenced by biomass burning as the interannual variability of this emissions source is substantially high (Tsimpidi et al., 2016).'

RC17: 'Figure 4 and Figure 13: What is the averaging for the model and observations used to get these points? Was any seasonal analysis conducted? Are these points a mix of different seasons?'

> AC17: The duration which these measurements spans vary from a few days up to one year, with the majority being less than one month. Yes, these observations span different seasons. Only a handful of the campaigns were conducted in the same region across multiple different seasons. Because of this, seasonal analysis can be very misleading and that is why we have chosen to categorise by environment type and continent.

RC18: 'Page 14: Please add some explanation in the paragraphs below or elsewhere in the paper about how this work might differ between past work. Not necessarily in overall magnitudes, but in approach. For example, this work uses SOA precursors that

irreversibly form SOA, while past work has used SOA precursors that reversibly form SOA (e.g., volatility basis set). Explain how this might affect the results in this work especially the impact of wet/dry deposition?'

> AC: We thank the referee for pointing this out. We've added some description of important similarities/differences between this study and previous studies
>
> Page 15 lines 10-13: *Until now, the impacts of precursor deposition on SOA concentrations have only been quantified over Europe (Bessagnet et al., 2010) and North America (Knote et al., 2015), both of which using regional scale models, and treat SOA as semi-volatile. Note, Bessagnet et al. (2010) treat SOA formation by a single-step oxidation of parent VOC followed by reversible condensation into the aerosol phase. Knote et al. (2015) treat SOA formation using the VBS scheme.*

RC19: 'Section 5: This section would be much more effective if it were written more concisely.'

> AC19: We thank the referee for this feedback. We agree that this section is lengthy, but with so many simulations we feel it is important to steadily guide the reader through. However, if the referee has suggestions for specific sections of this text that could be removed then please let us know.

RC20: 'Page 19 line 12: See general comment above. The use of multigenerational here and throughout the work is misleading. I would suggest phrasing this instead as RO2 fate.'

> AC20: We completely agree with the referee here. Please refer back to RC #4 where we have responded to this comment.

RC21: 'Page 20 line 18: The lifetime of the RO2 radical being ~1day is quite unexpected. Please confirm this and considering the actual RO2 radical lifetime, rephrase this section.'

> AC21: Please refer back to RC #4 where we have responded to this comment.

RC22: 'Page 21 line 6: Is NO actually high in the Amazon in your model? It looks low in Figure 11.'

> AC22: We agree with the referee here and have removed 'Amazon' from this sentence.
>
> Page 21 lines 22-24: *At the surface, the highest annual-average surface NO concentrations (1-23 ppb(v)) are simulated over industrialised and urban regions of North America, China and Europe (Figure 11 a).*

RC23: 'Page 23 line 14: Increasing the molar SOA yield is not equivalent to changes in volatility. Please rephrase this and all paragraphs in this discussion. Unless you change the volatility of the SOA precursors and have SOA form reversibly you are not actually accounting for the changes in volatility.'

AC23: We completely agree with the referee here and have revised this text in the updated manuscript.

Page 23 line 31 to page 24 line 1: *Hence in a further simulation, the difference in fragmentation/functionalization between products of different peroxy radical oxidation pathways are accounted for, whereby the yield for the $RO_2+HO_2$ reaction is increased from 13 to 66 %, whilst the yield for the $RO_2+NO$ reaction is left at 13 % (Multi_nap_yield; Table 3).*

RC24: 'Page 23 line 16: See above, please reconsider/clarify why 0.66 was chosen as the SOA molar yield?'

AC24: We thank the referee for highlighting how the test cases were not adequately explained in the original manuscript. We have responded to this question above in RC15.

RC25: 'Page 29 line 27: Please expand on this paragraph.'

AC25: We thank the referee for this suggestion. We have revised and expanded this section in the updated manuscript.

Page 30 lines 25-28: *Additional simulations could reach even wider bounds on the global SOA budget. For instance, neglecting SOA precursor deposition combined with $VOC_{ANT/BB}$ undergoing oxidation with $NO/HO_2$-dependent yields would results in even higher global SOA production rates. These results suggest that both oxidation and deposition remain significant contributors to uncertainty in the global SOA budget.*

RC26: 'Table 2: Please explain how the field studies were used to derive these surface resistance values. What would these surface resistances be, if the henry's law constants used for wet deposition were used here instead?'

AC26: We thank the referee for this question. Just to clarify, we did not use the field studies to derive surface resistances from. Instead, the field studies themselves derived the surface resistances, which we use as inputs into our model. We have edited the caption for this table to be more clear about this. Unfortunately, we are unaware of what the corresponding Henry's Law constants would be for these values of surface resistances.

Page 45 lines 4-6; *Surface resistances for SOA precursors over the 9 different surface types in the model. 'Low' represents surface resistances of ROOH, which are taken field studies (Hall et al., 1999;Nguyen et al., 2015). 'High' represents surface resistances of CO.*

RC27: 'NOx sometimes has x subscripted and sometimes not. -You have 2 section 2.5 - There are a number of spelling errors throughout as noted below: -page 2, line 4 (improvements), line 6 (observations) -page 6, line 13 (precursors) -page 15, line 33 (respectively) -page 18 line 12 (translated) -page 26 line 26 (respectively) -page 26 line 28 (chemistry) -page 27 line 3 (Africa)'

AC27: We thank the referee for taking the time to notify of these mistakes, which have been corrected in the updated manuscript.

**Referee #2 (received on 19[th] November 2018)**

RC1: 'This work by Kelly et al. investigate the impacts of VOC deposition and oxidation mechanisms on SOA formation within the United Kingdom Chemistry and Aerosol (UKCA) model. This work evaluated simulated OA/SOA with surface and aircraft observed data in different areas around the globe. This work highlights the uncertainties in the global SOA budget associated with the changes in SOA schemes. I will suggest to accept this manuscript after minor revisions. My specific comments are listed below.'

> AC1: We thank the reviewer for their positive feedback on the manuscript.

RC2: 'Emissions. How may VOC from biomass burning and anthropogenic sources respectively? Are both VOCBB and VOCANT are assumed to emit from the surface? The biomass burning source could be elevate emissions and might be impact on some of the results in this study. For example, in page 21, " At higher levels, NO/HO2 reduces, suggesting an increasing importance of the HO2 pathway at higher altitudes. However, due to the fast chemical reactivity, the majority of SOA production occurs at the surface. For the majority of the atmosphere, the difference in the magnitudes of the oxidant concentrations favours the RO2 +NO pathway over the RO2 +HO2 pathway." Therefore, if the SOA production occurs in higher altitude because of elevate emissions, more SOA will produce through HO2.'

> AC2: This is a really interesting point. Firstly, both anthropogenic and biomass burning VOCs are emitted at the surface, despite our knowledge that some of these sources may be emitted at high altitudes (e.g. biomass burning and chimney stacks). Considering high altitude VOC emissions is something we would definitely like to do as the model continues to be developed. And we agree with the referee's comment that by doing this, the fate of the peroxy radical would be altered. We have added this to our discussion.
>
> Page 21 line 33 to page 22 line 1: *High altitude emissions of VOCs from biomass burning plumes may be more susceptible to forming $RO_2$ which react with $HO_2$. However, in this study, all $VOC_{ANT/BB}$ are emitted at the surface.*

RC3: 'Default Treatment of SOA. The SOG condenses irreversibly to form SOA in UKCA. Will it lead to a different result if the model assumes the SOG condenses reversibly to form SOA?'

> AC3: The volatility of SOA remains a highly disputed area. Repeating these simulations under a semi-volatile treatment of SOA may indeed affect the conclusions drawn. Overall though, it's very difficult to predict how the results would be affected.

RC4: 'Could author discuss about the potential impacts of the precursors deposition on SOA production associated with different source types?'

> AC4: This is really interesting question which we did not explore in the original manuscript. We have re-analysed the model output and calculated the relative contributions of biogenic versus $VOC_{ANT/BB}$ to global SOA production, and how this is affected by including deposition.
>
> Page 14 lines 25-29: *Prior to including deposition of SOA precursors, biogenic VOCs account for 57 % of the global annual-total SOA production rate, with $VOC_{ANT/BB}$ accounting for the remaining 43 %. By including deposition of SOA precursors, the relative importance of biogenic VOCs to global SOA increase; considering deposition of SOA precursors, biogenic VOCs account for 62 % of the global annual-total SOA production rate, with $VOC_{ANT/BB}$ accounting for the remaining 38 %. Hence, biogenic VOCs appear to be less susceptible to deposition than anthropogenic and biomass burning VOCs.*

RC5: 'Page 8 Line 27: "All carbonaceous primary emissions are emitted into the insoluble mode and transferred into the insoluble" I cannot understand this sentence.'

> AC5: This was also raised by Referee #1. We have revised this section of text.
>
> Page 7 lines 27-32; *Aerosol microphysical processes included are nucleation, coagulation, condensation, condensation ageing, hygroscopic growth and cloud processing. Species such as POA and BC are assumed to be emitted in insoluble forms. Condensation ageing refers to soluble vapours condensing on these insoluble POA and BC particles, and thus, rendering them soluble. No aging processes are applied to SOA.*

RC6: 'Page 12 Line 15-20. "Consequently, a stoichiometric yield of 66 % corresponds to a mass yield of 100%. Therefore, 66% is the highest stoichiometric yield that ensures conservation of mass without the addition of other atoms, such as oxygen." Why the mass conservation used here?'

> AC6: This is an important point the referee raises. The yields assigned to the different $RO_2$ pathways are highly uncertain. Firstly, the laboratory derived yields vary from one study to another, and are dependent on a variety of conditions. Secondly, $RO_2$ in this study is a lumped species, representing the peroxy radcials formed from a mixture of VOCs from both anthropogenic and biomass burning source. Consequently, selecting laboratory defined yields for $RO_2$ is challenging. Therefore, the objective of this study is to explore the impacts of a low/high yield pathway. So we apply a high yield to the $RO_2+HO_2$ pathway, by increasing it by a factor of 5 (from 13 to 66 %). This 66 % stoichiometric yield happens to correspond to 100 % mass yield.

RC7: 'Page 14. Line 20. "all VOC source ranging from 47 to 74 Tg (SOA) a-1" change to "47 to 75"?'

> AC7: We thank the referee for pointing this out.

RC8: 'Page 15. Line 17-20. What is the lifetime of SOA in this study?'

> AC8: We thank the referee for pointing out that we didn't state the SOA lifetime in the original manuscript. The global-average annual-average SOA lifetime varies from 4 to 5 days across the simulation conducted in this study.
>
> Page 15 lines 31-32: *Across these simulations where the deposition of SOA precursors is altered, the global-average annual-average SOA lifetime varies from 4.3 to 4.7 days (not shown).*
>
> Page 26 lines 31-32: *Across these simulations where the $VOC_{ANT/BB}$ oxidation scheme is varied, the global-average annual-average SOA lifetime varies from 4.4 to 5.0 days (not shown).*

RC9: 'Page 15. Line 30. How model predicted SOA compare to the observations? Do they use monthly averaged or median values? Since "1.875◦ longitude by 1.25◦ latitude" in this study is a really coarse resolution, will the comparison with remote sites seems better?'

> AC9: We thank the referee for this important question. We have added some description on how the observed and simulated concentrations are compared.
>
> Page 16 lines 7-10: *Observed SOA concentrations are in the form of averages over the campaign period (which ranges from a few days to one year), and span from 2000 to 2010. This observed concentrations are then matched to the grid box which they fall in, with the simulated monthly averages being selected for the year 2000. Hence, there is a mismatch in terms of the measurement year and the simulated year.*

RC10: 'Page 19. Line 25-30. " these changes in annual-total VOCANT/BB oxidation rates within emissions source regions correspond to reductions between 10 and 30 % (not shown). By contrast, downwind of many emissions source regions, the lower reactivity acts to enhance VOCANT/BB oxidation rates." It is really hard for me to find the down- wind emission source regions because the largest increase occurs in source regions such as China and East US. Could the author give a map plot to point out where these downwind regions are?'

> AC10: We appreciate that these changes are quite difficult for the reader to see. We have changed the language to emphasise that these changes are small. With regards to an additional figure, this is a good suggestion. However, The manuscript is already quite long and we would prefer not to add anymore figures.

RC11: 'Page 26. Line 5-10. " Although the global annual-total SOA production rates are identical, the global annual-average SOA burden is 10 % greater when using benzene as the parent VOC undergoing multi-generational oxidation, highlighting the strong spatial gradients in SOA lifetime." Could author explain how the SOA lifetime changes?'

AC11: This is a really interesting point which would like to explain further. We believe that the SOA lifetimes varies throughout the model, both in the horizontal and vertical extent. For instance, SOA in the lower layers of the model is susceptible to wet removal whereas SOA above clouds is not. For this reason, by delaying VOC oxidation through reductions in parent VOC reactivity, SOA is produced at higher altitudes. Therefore, slowing down VOC reactivity across this series of aromatic compounds results in a lengthening of the SOA a longer lifetime. Unfortunately, this version of UKCA does not have spatially resolved SOA lifetime diagnostics, only a global. Therefore, whilst we have a theory on why the SOA lifetime is changing across these simulations, without evidence from spatially resolved SOA lifetime diagnostics, we would prefer not go into too much detail, as this theory is unsubstantiated.

**Referee #3 (received on 29[th] November 2018)**

RC1: 'Secondary organic aerosol (SOA) is an important but the least understood component of atmospheric aerosols. SOA life-cycle involves many chemical and physical processes, including emission, gas-phase chemistry, aqueous/solid phase chemistry, condensation, deposition and etc. This makes the global SOA modeling really challenging. This manuscript investigated the sensitivities of SOA formation to the different volatile organic compound (VOC) deposition and oxidation mechanism use a global chemistry-climate model (UKCA). It also compared these sensitivity simulations against the observations to see how these difference mechanisms affect the model- observation agreements. Overall, this manuscript is organized well and provide readers deep insights on how VOC deposition and oxidation reactions affect the SOA production. I recommend publishing it after the authors address my comments below.'

AC1: We thank the referee for taking the time to review this manuscript.

RC2: 'It is not clear to me why the authors only use aromatics as the biomass burning SOA precursor. How representative are the aromatics for the biomass burning SOA precursor?'

AC2: We thank the referee for this question. As noted in the original manuscript, aromatic compounds represent only a minor fraction of biomass burning (and anthropogenic) VOC emissions on the global scale (see Page 29 lines 16-18). However, from the perspective of SOA formation from these emissions sources, aromatic compounds have received the widest attention from laboratory and field studies. Hence, by selecting aromatic compounds as surrogate species to represent out SOA formation from biomass burning and anthropogenic VOC source, we are able to use the wealth of published data on these compounds, including oxidation mechanisms, reaction yields, and reaction kinetics. Within the conclusion, we've added a note to clarify that aromatic emissions are a minor component of anthropogenic and biomass burning emissions.

Page 30 lines 13-15: *Note however, that aromatic compound emissions represent only a minor fraction of the global annual-total VOC$_{ANT/BB}$ emission rate, which is 176 Tg (VOC$_{ANT/BB}$) a$^{-1}$.*

RC3: 'All model simulations underestimate the observed OA concentrations. The authors should at least discuss the reasons for this underestimation and its potential impact on this paper's conclusions.'

AC3: We thank the referee for this comment. This is a very interesting point that was not explored fully in the original manuscript. We have added a paragraph to the conclusion.

Page 30 lines 15-26: *In this study, observed OA/SOA concentrations generally exceed simulated OA/SOA concentrations. This is true at the surface and throughout the boundary layer. This model negative bias is very likely due to missing SOA (a) S/IVOC emissions, and (b) aqueous phase SOA production. As a result of these missing SOA source, care should be given when drawing conclusions on how variations in VOC deposition and oxidation mechanisms impact model agreement with observations. For instance, this study begins with a model negative bias, whereby inclusion of SOA precursor deposition worsens the model negative bias. However, if this study were to include S/IVOC emissions and aqueous phase SOA production, it would be possible to begin these series with a positive model bias. If this was the case, the inclusion of SOA precursor deposition would reduce the model positive bias. This study conclusively demonstrates that variations in VOC deposition and oxidation mechanisms do indeed alter the agreement between model and observed OA/SOA concentrations. However, as the sign of the model bias (i.e. positive or negative) could be sensitive to which SOA source are included, this study does not conclusively demonstrate if these model updates lead to an improvement or worsening of model agreement with observations.*

RC4: 'P3, line 1. Kelly et al., 2018 is not listed in the reference list.'

AC4: We thank the referee for pointing this out. The citation has been added to the reference list in the updated manuscript.

RC5: 'P3, line 7. Suggesting changing "an aspects of SOA" to "another aspect of SOA", because the previous sentence already described one aspect of SOA difference between different models.'

AC5: We thank the referee for this recommendation, which has been included in the updated manuscript.

RC6: 'P4, line 8-9. Can you list some references to support this statement?'

AC6: We thank the referee for this comment. We have revised the text.

Page 2 lines 1-3: *The first studies to quantify the SOA yields from aromatic compounds (Odum et al., 1997;Odum et al., 1996) are not high enough to account for the concentrations of aromatic SOA observed in field studies*

*(Tsigaridis and Kanakidou, 2003;Hoyle et al., 2007). For instance, early estimates…*

RC7: 'P8, line 20. Section 2.4 should be section 2.3. And also please change the section 2.5 number.'

AC7: We thank the referee for notifying us of these typos. These have been fixed in the updated manuscript.

RC8: 'P8, line 27. Please change VOCBB and VOCANT to "VOCBB" and "VOCANT" to be consistent with the rest of paper.'

AC8: Thank you for pointing out these typos.

RC9: 'P8, line 27. The second "insoluble" should be "soluble".'

AC9: We thank the referee for pointing out this typo. This text has now been revised according to another referee's comment.

RC10: 'P9, line 24-26. So the model includes both the isoprene oxidation that leads to SOA formation and the isoprene oxidation in the Mainz Isoprene Mechanism? Isn't this double counting isoprene oxidation?

AC10: This is really interesting question. We have looked back over the oxidation mechanism and have realised an error. For these simulations, and those described in Kelly et al. (2018), isoprene oxidation is split over multiple parallel oxidation reactions. This is due to there being a maximum limit of products per reaction in UKCA. Therefore, to capture all the products of isoprene oxidation from the Mainz Isoprene Mechanism (MIM), and the products which are assumed to go on to form SOA (here, SOG), isoprene oxidation is split over 3 reactions. Isoprene is emitted at around 500 Tg (isoprene) $a^{-1}$. But in the atmosphere, isoprene reacts under multiple oxidation reactions. With only 70 Tg (isoprene) reacting through the SOA production channel, after applying the 13 % molar yield, this results in a global SOA production rate from this source of only 20 Tg (SOA) $a^{-1}$. So isoprene isn't being double counted, but rather 'under counted'. Under this mechanism, the overall yield of SOA (20 Tg (SOA) $a^{-1}$) from isoprene oxidation, is around 4 %, instead of 13 % which was quoted in the manuscript. In Kelly et al. (2018), this complication in isoprene oxidation was not explained, and we are in the process of applying an erratum/corrigendum to that paper. As this current paper is mainly focussed on the anthropogenic and biomass burning SOA sources, we believe that this error would have a minor effect on this paper.

RC11: 'P11, line 20. Can the authors briefly describe the kinetics for aromatic oxidations here? So the readers don't have to read the table when reading the text.'

AC11: This is a good idea. We have provided the rate coefficients within the main body of text in the updated manuscript.

Page 11 *lines 22-26: At 298 K, the rate coefficients for the reaction of OH with naphthalene, toluene and benzene are 23.2, 5.62, and 1.22 $\times 10^{-12}$ cm$^3$ molecule$^{-1}$ s$^{-1}$, respectively (Table 1). At 298 K, the rate coefficients for the reactions of the peroxy radical with HO$_2$ and NO are 14.7 and 8.42 $\times 10^{-12}$ cm$^3$ molecule$^{-1}$ s$^{-1}$, respectively (Table 1). Note, these rate coefficients are used for the peroxy radical irrespective of the identity of the parent VOC (naphthalene, toluene or benzene).*

RC12: 'P12, line 18. "Different molecular masses". What molecular weights are used for RO2 and SOG in the model. SOG is a lumped species, right? So how do the authors know the molecular weight of SOG?'

AC12: We thank for referee for this interesting question. The referee is correct, both RO2 and SOG are lumped species and we do not know the relative molecular masses. We assume that the RO2 intermediate has an identical relative molecular mass to the parent hydrocarbon (VOC$_{ANT/BB}$) of 100 g mol$^{-1}$. SOG, which existed in the model before VOC$_{ANT/BB}$ was included, has a relative molecular mass of 150 g mol$^{-1}$.

RC13: 'P13, line 20-25. Did the authors account for the seasonal variation of biomass burning VOC emissions in the model (i.e. monthly change emissions)?'

AC13: Yes, we account for seasonal variation in biomass burning VOC emissions. We have added 'monthly-mean' to our description of how the VOC$_{ANT/BB}$ emissions are calculated.

Page 8 line 11; *monthly-mean*

RC14: 'P18, line 15. OH can be indirectly constrained by the CH4 lifetime.'

AC14: We thank the referee for noting this and we have added it to the updated manuscript.

Page 17 lines 1-2: *Alternatively, the OH concentration can be constrained indirectly from the CH$_4$ lifetime. Overall, the OH concentration is a difficulty quantity to capture in a global model.*

RC15: 'P24, line 24. "Favors the likelihood of RO2 radicals entering the high-yield HO2 path- way". Why? I don't understand the reason for that.'

AC15: Similar emissions patterns between VOC$_{ANT/BB}$ and NOx means that if RO$_2$ is generated quickly, the radical has a high probability of then reacting with NO. Reducing the reactivity of VOC$_{ANT/BB}$ delays VOC$_{ANT/BB}$ oxidation, such that RO$_2$ is formed away from the emissions source where NOx concentrations are lower and the probability of entering the NO pathways are reduced. We have revised to text as follows.

Page 25 lines 11-13: *Reducing the chemical reactivity of VOC$_{ANT/BB}$ reduces the global oxidation rate, whilst at the same time, favours the likelihood of*

*RO₂ radicals entering the HO₂ pathway (which has a higher SOA yield than the NO pathway).*

RC16: 'P26, line 30. "Figure 12", is it meant to be Figure 13?'

AC16: Yes, thank you for pointing out this mistake, which has been rectified in the updated manuscript.

RC17: 'P27, line 11-17. This argument is confusing to me. Can the authors elaborate that?'

AC17: We thank the referee for notifying us on this confusing section of text. We are trying to explain to the reader that the aircraft campaigns used in this study are classified (by themselves) as being conducted in polluted or biomass burning influenced regions (of Europe and Asia, respectively). Yet global emissions inventories and global models (like this study) would indicate Africa and South America as biomass burning hotspots, and perhaps Asia as a polluted hotspot. So these aircraft campaigns have a serious lack of geographical coverage, and are perhaps not indicative of polluted or biomass burning influenced regions.

RC18: 'Reference list. There are some references with titles being all capital letters. Please change them.'

AC18: We thank the referee for pointing this out. The references have been corrected in the updated manuscript.

RC19: 'Tabe 1. Please add the VOCANT and VOCBB oxidation kinetics in the "existing reaction kinetics" subsection.'

AC19: We do not explicitly have oxidation kinetics for $VOC_{ANT/BB}$, but instead vary from existing reactions (e.g. naphthalene, toluene, monoterpene, etc.), which are included in this table. We have added a sentence on this to the caption of Table 1 for clarity.

[revised manuscript text omitted]